

# Controlling factors on the global distribution of a representative marine heterotrophic diazotroph phylotype (Gamma A)

Zhibo Shao[1], Ya-Wei Luo[1]

[1]State Key Laboratory of Marine Environmental Science and College of Ocean and Earth Sciences, Xiamen University, 361102
Xiamen, Fujian, China

*Correspondence to*: Ya-Wei Luo (ywluo@xmu.edu.cn)

**Abstract.** Heterotrophic diazotrophs emerge as a potentially important contributor to the global marine $N_2$ fixation, while the factors controlling their distribution are unclear. Here, we explored what controls the distribution of the most sampled heterotrophic diazotroph phylotype, Gamma A, in the global ocean. First, we analyzed the relationship between *nifH*-based Gamma A abundance and climatological biological and environmental conditions. The carrying capacity of Gamma A abundance increased with net primary production (NPP) and saturated when NPP reached ~400 mg C $m^{-2}$ $d^{-1}$. The reduction in Gamma A abundance from its carrying capacity was mostly related to low temperature, which possibly slowed the decomposition of organic matter, and high concentration of dissolved iron, to which the explanation was elusive but could result from the competition with autotrophic diazotrophs. Using a generalized additive model, these climatological factors together explained 41% of the variance in the Gamma A abundance. Second, in additional to the climatological background, we found that mesoscale cyclonic eddies can substantially elevate Gamma A abundance, implying that Gamma A can respond to short-term features and benefit from stimulated primary production by nutrient inputs. Overall, our results suggest that the distribution of Gamma A is most likely determined by the supply of organic matters, not by those factors controlling autotrophic diazotrophs, and therefore insight a niche differentiation between the heterotrophic and autotrophic $N_2$ fixation. More samplings on Gamma A and other heterotrophic diazotroph phylotypes are needed to better reveal the controlling mechanisms of heterotrophic $N_2$ fixation in the ocean.

## 1 Introduction

Dinitrogen ($N_2$) fixation, mostly conducted by prokaryotic bacteria (termed "diazotrophs"), is an important bioavailable nitrogen (N) source to the ocean (Moore et al., 2018; Karl et al., 2002). Although autotrophic cyanobacteria have been recognized as important diazotrophs in the ocean (Zehr, 2011), heterotrophic non-cyanobacteria diazotrophs (NCDs) have been widely detected (Moisander et al., 2017; Riemann et al., 2010) and sometimes even found to dominate the diazotrophic gene pools in surface oceans (Farnelid et al., 2011). For example, NCD *nifH* (a gene encoding subunit of nitrogenase enzyme) amplicons were far superior in number to autotrophic diazotrophs at some sampling sites in the South Pacific Ocean (Halm et al., 2012; Moisander et al., 2014), Indian Ocean (Shiozaki et al., 2014; Wu et al., 2019) and South China Sea (Ding et al.,





2021). Metagenomic studies also revealed the dominant presence of diverse $N_2$-fixing proteobacteria in ocean genomic databases (Delmont et al., 2018; Delmont et al., 2021). Additionally, *nifH* of NCDs was also detected in subphotic seawaters (Benavides et al., 2018a) and oxygen deplete zones (Jayakumar and Ward, 2020; Loescher et al., 2014) where nitrogen loss was considered heavy (Lam and Kuypers, 2011). Although the $N_2$ fixed by heterotrophic diazotrophs is still not well quantified, substantial $N_2$ fixation found in aphotic zones (Rahav et al., 2013; Bonnet et al., 2013) and in experiments with photosynthetic

inhibitors (Rahav et al., 2015; Geisler et al., 2020), as well as recovered transcripts of the NCD *nifH* gene (Fernandez et al., 2011; Gradoville et al., 2017), supports active heterotrophic $N_2$ fixation in the ocean.

It is unclear what controls the growth and distribution of NCDs, as most of them are uncultivated (Bombar et al., 2016). Apparently, NCDs are different from their autotrophic counterparts in depending organic matter as their carbon and energy sources, which can be supported by experimental evidence that $N_2$ fixation is stimulated by adding dissolved organic matter

(DOM) (Rahav et al., 2016; Rahav et al., 2015; Bonnet et al., 2013; Bentzon-Tilia et al., 2015). However, other studies showed contradictory results in which DOM addition did not always stimulate the activity of NCDs (Benavides et al., 2018b; Benavides et al., 2015). Due to sensitivity to $O_2$ and high energy requirements of $N_2$ fixation (Bombar et al., 2016), abundant NCDs were found to attach to particles which supposably provide the diazotrophs a microenvironment with depleted oxygen and rich organic matter (Riemann et al., 2010; Farnelid et al., 2010; Scavotto et al., 2015; Pedersen et al., 2018; Geisler et al., 2019).

NCDs were also detected in diatom mats (Martínez et al., 1983), probably releasing their fixed N to support diatom growth (Bonnet et al., 2016). An isolated strain of diazotrophic Alphabacteria from Baltic Sea found to equip with photosynthetic genes (Bentzon-Tilia et al., 2015) may complicate this issue, doubting whether NCDs can be mixotrophic and also depend on light.

Although dissolved inorganic nitrogen (DIN) is generally considered to inhibit marine $N_2$ fixation (Karl et al., 2002; Zehr and

Kudela, 2011), NCDs are active in DIN-replete environments such as estuaries (Geisler et al., 2020), coastal zones (Li et al., 2020), upwelling regions (Geisler et al., 2020; Moreira-Coello et al., 2017; Dekaezemacker et al., 2013) and other eutrophic seas (Bird and Wyman, 2013). Culture experiments showed that the DIN inhibition effect on NCDs can be strain specific (Bentzon-Tilia et al., 2015; Martínez-Pérez et al., 2018). Temperature could be another factor controlling NCDs which may prefer warm oligotrophic surface oceans (Langlois et al., 2015; Shiozaki et al., 2017), the same region where the majority of

autotrophic $N_2$ fixation supposably occurs (Wang et al., 2019; Luo et al., 2014). Similar to cyanobacterial diazotrophs, phosphate can also limit the growth of NCDs in oligotrophic environments (Rahav et al., 2015). Regarding other important factors that control autotrophic diazotrophs such as iron (Fe) and stratification, there have been, to our knowledge, no studies reporting its relationship with NCDs.

Mesoscale eddies can also impact NCD abundance. Although anticyclonic eddies were generally considered to benefit

autotrophic diazotrophs by inhibiting vertical DIN input from deep waters (Liu et al., 2020; Fong et al., 2008; Church et al., 2009), a class of NCDs, the Gammaproteobacteria, were found to dominate diazotrophic communities inside cyclonic eddies



in the South China Sea (Zhang et al., 2011; Liu et al., 2020). Different types of mesoscale eddies may have discrepancies in impacting the ecophysiology of NCDs and their autotrophic counterparts.

The major known NCD classes include bacteria such as Alphaproteobacteria, Gammaproteobacteria, Epsilonproteobacteria,

Betaproteobacteria, Firmicutes belonging to Cluster I of *nifH* clusters, and some obligate anaerobic bacteria and archaea belonging to Cluster III of *nifH* clusters (Zehr et al., 2003; Riemann et al., 2010). Among them, Gamma A is the most sampled and studied phylotype. Gamma A represents a part of uncultured Gammapreoteobacterial sequences isolated from the open ocean, and its cluster is distantly related to cultured NCD (Langlois et al., 2015). Gamma A *nifH* gene expression has been widely found in the global ocean, revealing its important role in marine $N_2$ fixation (Bird et al., 2005; Moisander et al., 2014;

Langlois et al., 2015; Shiozaki et al., 2017).

In this study, we collected, to our best knowledge, all the reported in situ measurements of Gamma A *nifH* copies. We then analysed the relationship between this *nifH*-based Gamma A abundance and the long-term background of ecological and environmental factors by using their climatological monthly averages. We further explored the short-term influence of mesoscale eddies on Gamma A abundance. Our analyses revealed that local primary productivity, temperature, dissolved Fe

concentration and the occurrence of cyclonic eddies can be the main factors impacting the distribution of Gamma A in the global ocean.

## 2 Methods

### 2.1 Data summary and quality control of Gamma A abundance

A total of 1690 *in situ* measurements of *nifH* copies of Gamma A in the Pacific, Atlantic and Indian Oceans were collected

from 18 published papers (Table 1), and are available in a data repository (https://doi.org/10.6084/m9.figshare.17284517) (Shao and Luo, 2021). Gamma A was sometimes also named γ-24774A11 in the collected papers (Moisander et al., 2008). All these data were measured using quantitative polymerase chain reaction (qPCR). Note that the primer of Gamma A used by Langlois et al. (2015) in the North Atlantic was slightly different from other studies. Most samples (87%) were collected in the surface 100 m seawater.

There were 616 data points reporting zero *nifH* copies which theoretically could indicate that they were below the detection limit. However, by assuming a log-normal distribution of the data, there should be only 72 data below the common detection limit of qPCR (100 copies $L^{-1}$) (Luo et al., 2012) (Fig. S1). Therefore, these zero-value data were more likely caused by primer specificity or insufficient volume of water samples, instead of representing the low abundance, and therefore were not used in our analyses.

Chauvenet's criterion was used to identify outliers by first log-transforming all the data (Glover et al., 2011). Two outliers (0.22 copies $L^{-1}$ and 0.33 copies $L^{-1}$) were removed because their probability of deviation from the mean was smaller than



1/(2*n*), where *n* was the number of data. Even though they can be reliable, we excluded them in the analyses to avoid possible biases.

In the following analyses, we represented Gamma A abundance using its *nifH* copies, although we noted that variations in *nifH*

copies in different diazotroph cells were reported (White et al., 2018; Sargent et al., 2016).

**Table 1. Data source of complied Gamma A *nifH* samples.**

| Reference | Location | Latitude | Longitude | Depth (m) |
|---|---|---|---|---|
| *Pacific Ocean* | | | | |
| Moisander et al. (2008) | South China Sea | 9°-12°N | 107°- 110°W | 0-1700 |
| Bombar et al. (2011) | Mekong River Plume | 9°-11°N | 106°- 107°W | 0 |
| Hamersley et al. (2011) | Southern California Bight | 33°N | 118°W | 5-885 |
| Moisander et al. (2014) | South Pacific Ocean | 15°- 30°S | 177°E- 155°W | 4-175 |
| Shiozaki et al. (2015) | Northwest Pacific | 38°- 39°N | 141°- 143°W | 0-119 |
| Berthelot et al. (2017) | Western Pacific Ocean | 3°- 12°S | 140°- 160°W | 0 |
| Shiozaki et al. (2017) | North Pacific Ocean | 0°- 68°N | 168°- 170°E | 0-157 |
| Shiozaki et al. (2018a) | South Pacific Ocean | 0°- 40°S | 170°- 100°W | 0-215 |
| Shiozaki et al. (2018b) | Kuroshio | 25°-33°N | 124°- 139°W | 0-5 |
| Chen et al. (2019a) | Western Pacific Ocean | 0° - 21°N | 110°- 159°W | 0-150 |
| Cheung et al. (2020) | North Pacific Ocean | 7°- 54°N | 139°E- 80°W | 5 |
| *Atlantic Ocean* | | | | |
| Langlois et al. (2008) | North Atlantic Ocean | 0° - 40°N | 10°- 70°W | 5-100 |
| Benavides et al. (2016) | North Atlantic Ocean | 0°-21°N | 15°-75°W | 0-150 |
| Martinez-Perez et al. (2016) | Tropical North Atlantic Ocean | 11°-15°N | 21°-60°E | 5-200 |
| Moreira-Coello et al. (2017) | Upwelling Region off NW Iberia | 6°-18°N | 18°-54°E | 0 |
| Moore et al. (2018) | Tropical Atlantic Ocean | 0°-21°N | 15°-55°W | 0 |
| *Indian Ocean* | | | | |
| Shiozaki et al. (2014) | Arabian Sea | 4°S–20°N | 65° –70° E | 0-86 |
| Wu et al. (2019) | Bay of Bengal | 4°S–10°N | 84° –96° E | 0-200 |



## 2.2 Environmental and ecological parameters

Monthly climatological environmental and ecological parameters were used as predictors for Gamma A abundance (Table 2).

Temperature and concentrations of nitrate, phosphate and silicate were the products of the World Ocean Atlas (WOA) 2018 (www.nodc.noaa.gov)(Locarnini et al., 2018; Garcia et al., 2019), and excess phosphate ($P^*$) was derived from concentrations of phosphate and nitrate based on the Redfield ratio ($P^*$ = [phosphate] - [nitrate]/16). Dissolved iron (Fe) concentrations were obtained from the Community Earth System Model – Biogeochemistry module (Misumi et al., 2014). Dissolved organic carbon concentration used a product estimated by artificial neural network (Roshan and Devries, 2017). Mixed layer depth (MLD)

was downloaded from Ifremer (http://www.ifremer.fr/) using the criterion that the potential density of water parcels at the depth was 0.03 kg m$^{-3}$ higher than that at the surface (De Boyer Montégut et al., 2004). Net primary production used a satellite data based on the VGPM algorithm (Behrenfeld and Falkowski, 1997) (http://sites.science.oregonstate.edu/ocean.productivity/). Surface photosynthetically active radiation (PAR) was downloaded from MODIS-Aqua program (http://oceancolor.gsfc.noaa.gov/). To estimate vertical profile of PAR, we first obtained the

estimated euphotic zone depth $Z_e$ (https://oceancolor.gsfc.nasa.gov/) at 1% surface PAR based on an inherent optical property (IOP)-centered approach (Lee et al., 2005), and used it to estimate the attenuation coefficient:

$$k_d = \frac{\ln(0.01)}{Z_e}. \tag{1}$$

The PAR at depth $z$ can be calculated while we assumed organisms in the mixed layer were exposed to PAR homogenously:

$$PAR(z) = \begin{cases} PAR_0 e^{-k_d z} & (z > MLD) \\ \frac{1}{MLD} \int_0^{MLD} PAR_0 e^{-k_d z} dz & (z < MLD) \end{cases}, \tag{2}$$

where $PAR_0$ is the surface PAR.

To identify if the Gamma A abundance is sampled in cyclonic or anticyclonic eddies, we used daily sea level anomaly data (SLA) provided by AVISO program (www.aviso.altimetry.fr). Only those cyclonic (negative SLA) and anticyclonic (positive SLA) eddies with clear shapes were recorded.

All the variables used in the analyses are available in a data repository (https://doi.org/10.6084/m9.figshare.17284517) (Shao

and Luo, 2021).



**Table 2. Environmental and ecological parameters.**

| Environmental parameters | Symbol | Source | Spatial resolution | Log-normally distributed and log-transformed |
|---|---|---|---|---|
| Temperature (℃) | T | | 1° | No |
| Dissolved inorganic nitrogen (μM) | DIN | World Ocean Atlas 2018 | 1° | Yes |
| Dissolved inorganic phosphate (μM) | DIP | | 1° | Yes |
| Excess P (μM) | P* | DIP-DIN/16 | 1° | No |
| Fe (nM) | Fe | CESM | 1° | Yes |
| Mixed layer depth (m) | MLD | IFREMER | 2° | Yes |
| Net primary production (mg C m⁻² d⁻¹) | NPP | VGPM | 1/6° | Yes |
| Photosynthetic active radiation (E m⁻² d⁻¹) | PAR | MODIS-Aqua | 1/6° | Yes |
| Dissolved organic carbon (μM) | DOC | Model | 1° | No |

### 2.3 Statistical analyses

For Gamma A data points sampled in the same months and the same depth bins (defined in WOA), they were binned to 2° × 2° grids to help eliminate possible biases caused by concentrated samplings in specific regions, resulting in 893 binned means of log-10 based Gamma A *nifH* abundance. The corresponding environmental and ecological parameters were also averaged
to the same bins when necessary. Univariate Pearson correlation was used to evaluate linear relationship between Gamma A abundance and each environmental or ecological variable.

We also used the generalized additive model (GAM) using R package 'mgcv' (Wood, 2017) to demonstrate non-linear relationships between the multiple predictors and the Gamma A abundance:

$$y = \alpha + \sum_{i=1}^{n} s(x_i) + \varepsilon, \tag{3}$$

where y is response variable (Gamma A abundance), $x_i$ is the $i$th predictor (i.e., the environmental or ecological variable), $\alpha$ is the intercept, $s(x_i)$ is a linear combination of smooth functions of predictor $x_i$, $n$ is number of predictors and $\varepsilon$ is standard error. To avoid over-fitting to noise, the Restricted Maximum Likelihood (REML) method was selected for the GAM smoothing parameters of every predictor with the basis function number (k) set to 9 (Wood et al., 2016). In the model selection of GAM, a double penalization approach was used to identify and remove those insignificant predictors with large smoothing
parameters and set them to zero functions (Marra and Wood, 2011).

The scientific color maps are used in several figures to prevent visual distortion of the data and exclusion of readers with color-vision deficiencies (Crameri et al., 2020).



# 3 Results and Discussion

## 3.1 Global distribution of Gamma A *nifH* abundance

The *nifH* gene abundance ranges from 1 to $10^7$ copies L$^{-1}$ in the global ocean and shows an approximately log-normal distribution (Fig. S1). High abundance of Gamma A *nifH* abundance over $10^5$ copies L$^{-1}$ is prevalent in subpolar North Pacific, tropical Atlantic and Bay of Bengal (Indian Ocean) (Fig. 1). Most Gamma A abundance data were sampled above 100 m, particularly in the upper 25 m. The deepest datum was sampled at 340 m in Southern California Bight (Hamersley et al., 2011). Available data showed that *nifH* abundance decreased with depth in Southwestern Pacific Ocean, but did not have an apparent

trend from the surface down to 200 m in the tropical Atlantic Ocean and the Indian Ocean (Fig. S2). More data particularly in deep waters are needed to better and more reliably reveal the vertical pattern of Gamma A abundance.

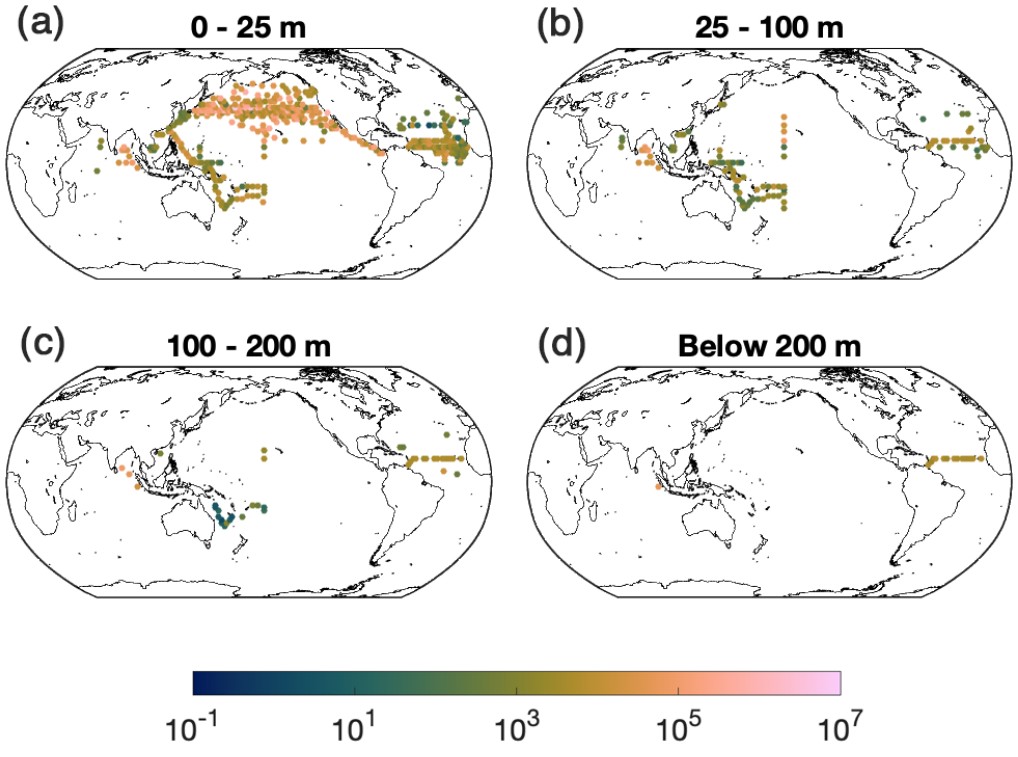

**Figure 1. Gamma A abundance (*nifH* copies L$^{-1}$).** The panels show data in depth ranges of (a) 0 – 25 m, (b) 25 – 100 m, (c) 100 – 200 m and (d) below 200 m. The data are binned to 2° × 2°.






**3.2 Primary production determines the carrying capacity of Gamma A abundance**

The logarithm of Gamma A *nifH* abundance positively correlated to the logarithm of net primary production (NPP) (correlation

= 0.21, $p < 0.01$) (Fig. 2). More importantly, the upper bound of Gamma A abundance increased with the NPP [$\log_{10}$(Gamma A) = 5.1 $\mathrm{Log}_{10}$NPP - 6.3] when NPP < $10^{2.6}$ ($\approx$400) mg C m$^{-2}$ d$^{-1}$, above which the upper bound of Gamma A abundance saturated at ~$10^7$ *nifH* copies L$^{-1}$ (Fig. 2).

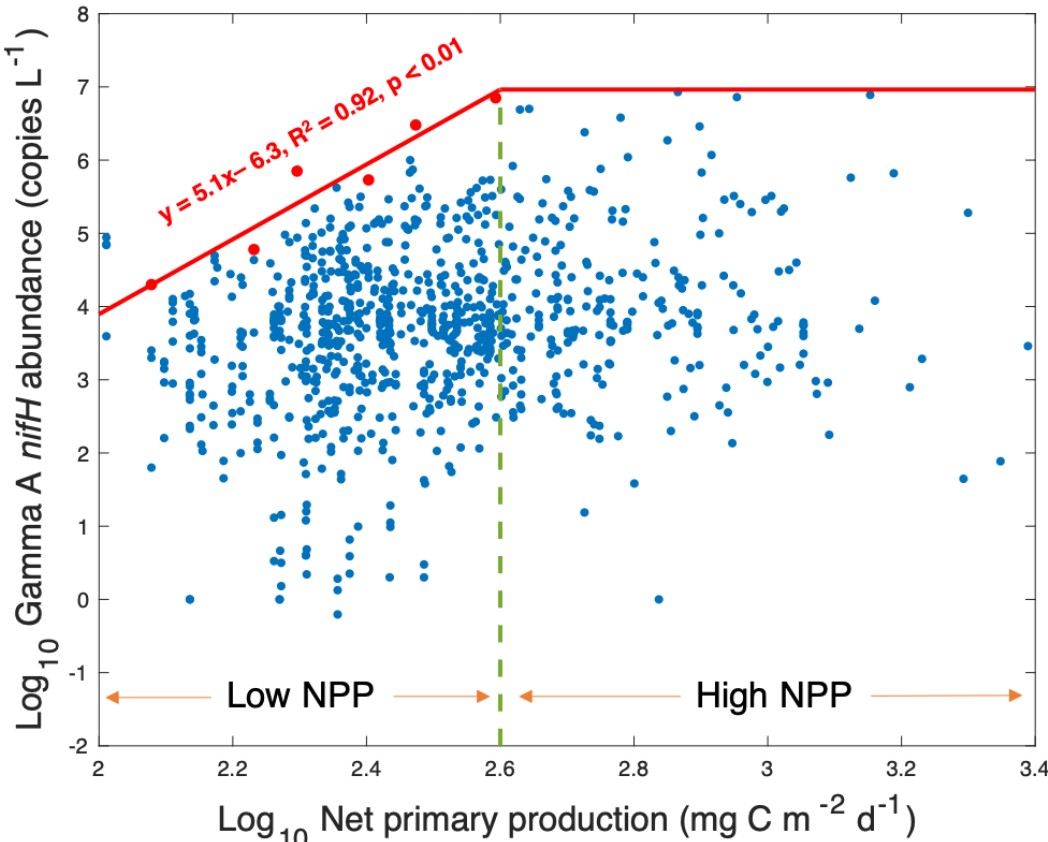

**Figure 2. The relationship between Gamma A abundance and net primary production.** Both Gamma A abundance and net primary production (NPP) are $\log_{10}$-transformed. The NPP-determined Gamma A carrying capacity (red line) in the "low" NPP range [< $10^{2.6}$ ($\approx$400) mg C m$^{-2}$ d$^{-1}$] is estimated by linearly fitting the highest Gamma A abundance values (red dots) in NPP intervals of $10^{0.1}$ mg C m$^{-2}$ d$^{-1}$. The Gamma A carrying capacity saturates at $10^{7.0}$ *nifH* copies L$^{-1}$ for NPP > $10^{2.6}$ mg C m$^{-2}$ d$^{-1}$ (the "high" NPP range).



These results indicated that local NPP could largely determine the carrying capacity of Gamma A abundance, which was expected because Gamma A was heterotrophic bacteria and needed sufficient supply of organic matter from primary producers, particularly for their energetically intensive $N_2$ fixation. This conclusion can also be partly supported by previous experimental studies in which the addition of organic carbon enhanced heterotrophic nitrogen fixation and NCD abundance in oligotrophic seas (Benavides et al., 2015; Rahav et al., 2016; Moisander et al., 2012; Dekaezemacker et al., 2013). Our finding contradicted

the hypothesis mentioned above that Gamma A preferred oligotrophic waters based on samples mainly in tropical and subtropical Pacific and Atlantic Oceans, in which Gamma A reached $8 \times 10^4$ *nifH* copies L$^{-1}$ (Shiozaki et al., 2018a; Langlois et al., 2015). However, the new dataset (Cheung et al., 2020) included in the present study showed even higher (over $10^5$ *nifH* copies L$^{-1}$) Gamma A abundance in the subarctic North Pacific (Fig. 1) where nutrient concentrations and NPP are generally high.

**3.3 Univariate linear relationships between environmental factors and Gamma A abundance**

We then analyzed what environmental factors may limit Gamma A abundance from reaching the carrying capacity. We first defined this disparity for each data point, $\Delta_{\text{Gamma-A}}$, as the observed Gamma A abundance minus corresponding carrying capacity in logarithmic space. That is, $\Delta_{\text{Gamma-A}}$ can be treated as the 'residual' of data to the carrying capacity line in Fig. 2. Therefore, a positive correlation between $\Delta_{\text{Gamma-A}}$ and an environmental factor can indicate that Gamma A prefers the increase

of this factor, and vice versa.

As the Gamma A carrying capacity saturated at an NPP of $10^{2.6}$ ($\approx$400) mg C m$^{-2}$ d$^{-1}$, we then divided our dataset into a low- and a high-NPP groups at this threshold (Fig. 2) in further analyses to address possibly different controlling factors and mechanisms on Gamma A abundance.

In the univariate linear analyses (Fig. 3), the most correlated variable to $\Delta_{\text{Gamma-A}}$ was the dissolved Fe concentration in both

groups, but surprisingly the relationship was negative. Temperature and DOC were positively correlated with $\Delta_{\text{Gamma-A}}$ in the low-NPP group, while the relationships turned negative when NPP was high. In terms of inorganic nutrients, $\Delta_{\text{Gamma-A}}$ correlated negatively to nitrate and correlated positively to the excess P (P$^*$) in the low-NPP group, while both relationships became insignificant in the high-NPP group. Phosphate had no significant relationship with $\Delta_{\text{Gamma-A}}$ in either group. Silicate was positively correlated to Gamma A, and the correlation became stronger in high NPP area. The positive correlation between

PAR and $\Delta_{\text{Gamma-A}}$, particularly when NPP is low, can be a result of a decreasing trend of Gamma A abundance with water depth. Lastly, $\Delta_{\text{Gamma-A}}$ and the mixed layer depth were negatively correlated only in the high-NPP group.





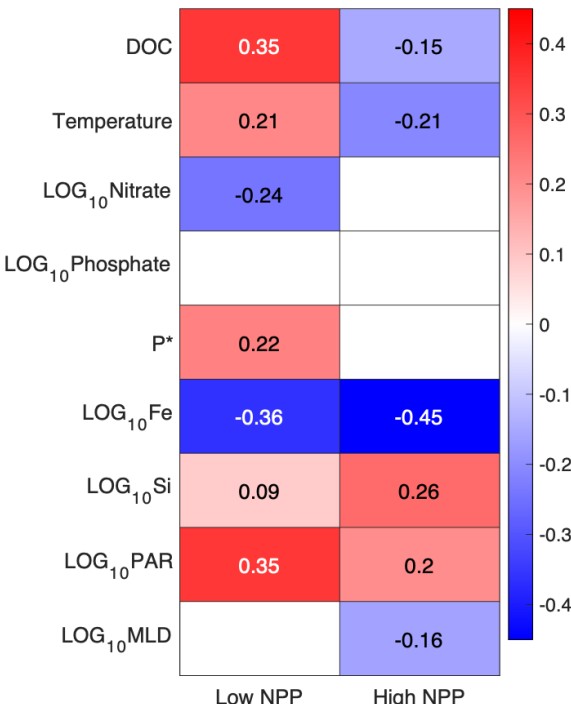

**Figure 3. Correlation between environmental factors and $\Delta_{\text{Gamma-A}}$.** The values are univariate Pearson correlation coefficients. Correlations with $p > 0.05$ are considered insignificant and left blank.

## 3.4. Multivariate nonlinear relationships between environmental factors and Gamma A abundance using GAM

Although the linear univariate correlations can provide basic information in analyzing relationships between each environmental factor and the Gamma A abundance, false relationships can be generated by intercorrelations existing among the environmental variables. Additionally, the relationships can also be nonlinear. We then used a GAM multivariate analysis to partly avoid these possible problems and to obtain a more reliable relationship (Fig. 4). Note that phosphate was not included in the GAM because it was not correlated to $\Delta_{\text{Gamma-A}}$ in both the low- and the high-NPP groups (Fig. 3) and its variance can nevertheless be partly represented by $P^*$.




**Figure 4. Partial effects of environmental variables on Δ<sub>Gamma-A</sub> using GAM multivariate analysis.** The analyses are conducted for the low-NPP group (a–h) and the high-NPP group (i–p), showing the anomaly of Δ$_{Gamma-A}$ contributed by the smooth function (blue line) and its 95% confidence interval (shadow) of each environmental variable. Data (circles) are shown as partial residuals after all other partial effects have been considered. The numbers in the parentheses of y-axis labels are the degree of freedom of the smooth functions. A degree of freedom smaller than 1 is equivalent to a linear line, and higher degrees of freedom represent more wiggly curves. The blue ticks on the x-axis also indicate the location of the data.



### 3.4.1 DOC

Although DOC was supposedly one of the major carbon sources for Gamma A, it did not impact $\Delta_{Gamma-A}$ in the low-NPP
group (Fig. 4a) and even showed a negative linear relationship with $\Delta_{Gamma-A}$ under high NPP (Fig. 4i). First, $\Delta_{Gamma-A}$ was the
residual to the NPP-determined carrying capacity and therefore was a collective indicator in which the effects of organic carbon
production had been largely removed. Additionally, low DOC concentrations in high-NPP regions may even indicate that the
DOC pool is more labile and can be more easily used (Jiao et al., 2014). Lastly, particulate organic matter (POM) can also fuel
Gamma A and can even create favorable oxygen-deplete microenvironments for Gamma A (Farnelid et al., 2019), but it was
not included in this study because of insufficient data.

### 3.4.2 Temperature

Temperature had a generally positive relationship with $\Delta_{Gamma-A}$ (Figs. 4b and 4j). This is consistent with several regional
studies in which a strong positive correlation between temperature and Gamma A abundance was also found (Shiozaki et al.,
2018a; Moisander et al., 2014). The relationship is expected considering the widely recognized increase in heterotrophic
bacterial production with temperature in the ocean because of stimulated bacterial metabolism (Kirchman and Rich, 1997;
Pomeroy and Wiebe, 2001). In addition, $\Delta_{Gamma-A}$ started to rise at a lower temperature (~15°C) in the high-NPP group (Fig.
4j) than that (~20°C) in the low-NPP group (Fig. 4b). The contribution of temperature to $\Delta_{Gamma-A}$ is larger in the low-NPP
group (Fig. 4b) than that in the high-NPP group (Fig. 4j). A possible reason is that the consumption rate of less labile DOC
produced in less productive regions is more sensitive to temperature (Lønborg et al., 2018; Brewer and Peltzer, 2017; Carlson
et al., 2004). This result implied that high temperature could be more important for Gamma A to activate accumulated semi-
labile DOC in low-productive oligotrophic oceans.

### 3.4.3 Nitrate and P$^*$

Neither nitrate nor P$^*$ had a substantial effect on $\Delta_{Gamma-A}$ (Figs. 4c–d and 4k–l). This is consistent with a previous review
showing that nitrate did not show an immediate inhibition of Gamma A (Moisander et al., 2017). How and to what extent
NCDs are inhibited by nitrate remains unknown (Bombar et al., 2016). Abundant Gamma A was found in oceans with high
nitrate concentrations (Bird and Wyman, 2013) or shallow nitracline (Shiozaki et al., 2014). The hypothesis that low-nitrate
and high-P$^*$ environments favor autotrophic diazotrophs is based on the competition of inorganic nutrients between the
diazotrophs and other phytoplankton (Karl and Letelier, 2008; Deutsch et al., 2007), while this competition should not occur
directly between NCDs and phytoplankton because the former is heterotrophic while the latter is autotrophic. Nevertheless,
high inorganic nutrients may still play a role in Gamma A distribution by indirectly impacting the carrying capacity of Gamma
A via NPP. If this is true, high nitrate then is a beneficial, instead of an inhibiting, factor on NCDs.



### 3.4.4 Iron

In both the low- and the high-NPP groups, $\Delta_{\text{Gamma-A}}$ generally showed a decreasing trend with the increasing dissolved Fe, except for a slight increase in $\Delta_{\text{Gamma-A}}$ when the dissolved Fe increased in the range of 0.01–0.1 nM (Figs. 4e and 4m). Our

dataset showed that high abundance of Gamma A was prevalently observed in the North Pacific Ocean (Fig. 1a), where Fe was considered as the dominant limiting factor for $N_2$ fixation (Sohm et al., 2011). Other Gammaproteobacterial phylotypes such as Gamma 3 and Gamma ETSP2 were also found to dominate the diazotrophic community in the eastern South Pacific (Turk-Kubo et al., 2014; Halm et al., 2012) where Fe heavily limited primary production (Knapp et al., 2016; Bonnet et al., 2008). It has also been suggested that Gamma A and unicellular cyanobacterial diazotroph UCYN-B share niches in Fe-

depleted western and southern Pacific Oceans (Moisander et al., 2014; Chen et al., 2019a), possibly to avoid competing with other Fe-demanding diazotrophs. Gammaproteobacterial diazotrophs may be equipped with siderophore releasing genes, such as that already reported in another versatile phylotype Gamma 4 (Cheung et al., 2021), and the released siderophores are an efficient tool in scavenging low-level Fe in the ocean (Boyd and Ellwood, 2010). Although more studies are certainly needed to further explore the ecological and physiological mechanisms and evolutionary reasons, the good survival of Gamma A in a

low-Fe environment is an intriguing finding that may expand our recognized space of active $N_2$ fixation in the ocean.

### 3.4.5 Silicate

Our GAM results also revealed a positive relationship between silicate and $\Delta_{\text{Gamma-A}}$ in both the low- and the high-NPP groups (Figs. 4f and 4n), indicating a possible association between Gamma A and diatoms. NCDs have been found on the surface of diatoms (Martínez et al., 1983) or on the diatom mats as discussed above. Diatom-dominant ecosystems tend to produce

abundant large particles either from dead diatoms and their aggregates or the fecal pellets generated by zooplankton (Tréguer et al., 2018). The large particles can be a good habitat for NCDs as already discussed. A metagenomic study also indicated the swimming motility gene expression of NCDs and their potential particle-attached lifestyle (Delmont et al., 2018). Our results then provide indirect evidence for this hypothesis.

### 3.4.6 Light

PAR did not show a substantial contribution to the variance of $\Delta_{\text{Gamma-A}}$ in our multivariate GAM analysis (Figs. 4g and 4o). The high correlation between PAR and $\Delta_{\text{Gamma-A}}$ (Fig. 3) was therefore more likely resulted from multicollinearity between PAR and other environmental factors. The decrease in Gamma A abundance with depth found in the Southwestern Pacific (Fig. S2a) and in other regions by previous studies (Moisander et al., 2008; Langlois et al., 2015; Chen et al., 2019b; Shiozaki et al., 2014; Wu et al., 2019) may therefore be attributed to higher productivity, more released photosynthetic products and

higher temperature in the surface ocean, instead of the direct effects of light such as the hypothesized photoheterotrophy of Gamma A (Moisander et al., 2014). The nearly constant Gamma A abundance with depth in the Tropical Atlantic Ocean and




the Indian Ocean and (Fig. 1 and Figs. S2b–c) can be the results of active transport of organic matter from the surface that fuels heterotrophic $N_2$ fixation in the dark deeper ocean.

### 3.4.7 Predictions based on GAM

Overall, the multivariate GAM model explained 45% and 39% of the variance in $\Delta_{\text{Gamma-A}}$ in the low- and high-NPP groups, respectively (Figs. 5a–b). The predicted $\Delta_{\text{Gamma-A}}$ generally followed the observed values, although it tended to underestimate the observed high $\Delta_{\text{Gamma-A}}$ (> -1) or overestimate the low $\Delta_{\text{Gamma-A}}$ (< -5) (Figs. 5a–b). The moderate explained variance indicated that although the tested environmental factors can substantially influence Gamma A abundance, there must be other untested factors and unknown mechanisms that can also substantially impact the Gamma A distribution.


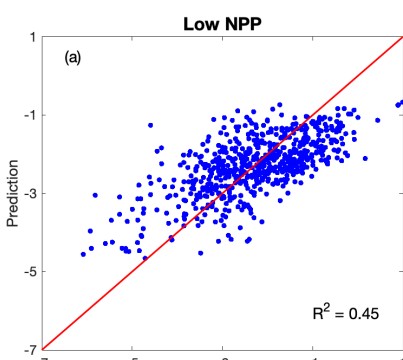
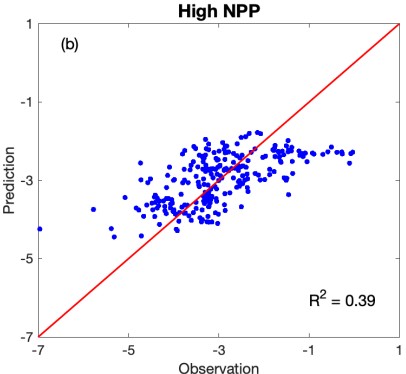
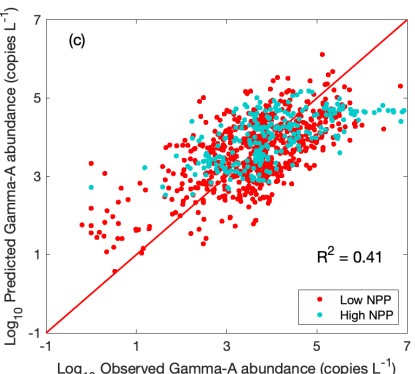

**Figure 5. Predictivity of GAM.** Predicted $\Delta_{\text{Gamma-A}}$ versus observed $\Delta_{\text{Gamma-A}}$ are shown in (a) the low-NPP and (b) the high-NPP data groups. (c) Comparison of predicted versus observed Gamma A *nifH* abundance. The red lines are 1:1 ratio of
prediction to observation.

The GAM models for $\Delta_{\text{Gamma-A}}$ (Figs. 5a–b) were added to the estimated NPP-based carrying capacity (Fig. 2) to form a prediction model for the Gamma A abundance (Fig. 5c). Although a substantial fraction of variance in Gamma A abundance was still unexplained ($R^2 = 0.41$), the predicted and the observed Gamma A abundances were generally consistent (Fig. 5c).
The predicted Gamma A abundance ranged mostly on the order of $10^1$–$10^6$ *nifH* copies L$^{-1}$, slightly narrower than that of the observations ($10^0$–$10^7$ *nifH* copies L$^{-1}$), which was mainly attributed to the performance of the GAM models for $\Delta_{\text{Gamma-A}}$ as discussed above.

Although the overall $R^2$ was at a moderate level of 41%, we applied this model to give a first-order estimate of Gamma A abundance in the surface ocean (Fig. 6a) from climatological NPP and environmental factors (Fig. S3), admitting that this
demonstration did not fully cover the observed spatial variance in Gamma A abundance. The results suggested that the Gamma



A was most abundant in the upwelling region in the Eastern Tropical South Pacific, the Southern Ocean and some coastal areas. The predicted high abundance in the Southern Ocean was mostly caused by its high nitrate concentration (Figs. S3g–h). However, the largest uncertainties for the predictions also exist in the Southern Ocean (Fig. 6b) as there were no Gamma A samples in this high-nitrate area (Fig. 1). Future sampling in the Southern Ocean can then test our predictions and reduce the

uncertainties. The prediction also showed the lowest Gamma A in the South Pacific Subtropical Gyre (Fig. 6a) where however the uncertainty was among the highest in the tropical and subtropical regions (Fig. 6b) similarly because it was under sampled (Fig. 1).

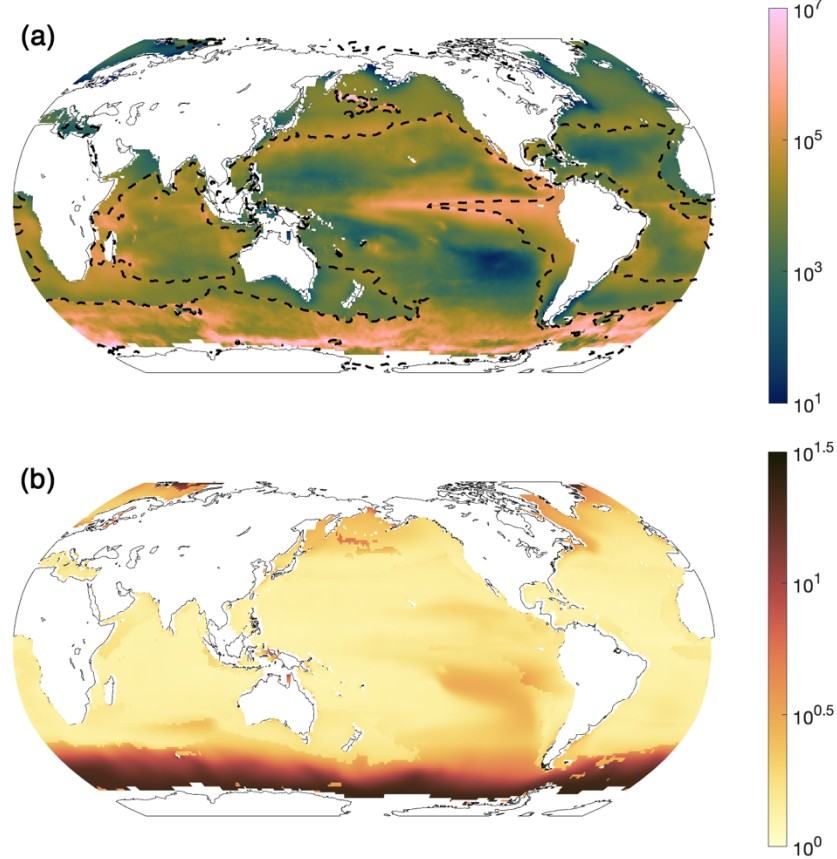

**Figure 6. Prediction of Gamma A abundance.** (a) Predicted annual mean surface (0–25 m) Gamma A abundance (*nifH* copies L$^{-1}$) and (b) the standard errors estimated by the GAM. Black dashed contours in (a) represent NPP of 10$^{2.6}$ (≈400) mg C m$^{-2}$ d$^{-1}$.





## 3.5 Impact of mesoscale eddies on Gamma A

The root-mean-square error (RMSE) of 0.86 and an $R^2$ of 41% in the prediction model (Fig. 5c) indicated that there was still
substantial unexplained variance in Gamma A abundance. One possible reason can be attributed to the climatological monthly
means we used in the environmental factors. We then explored whether the occurrence of short-term phenomena, such as the
mesoscale eddies, can impact Gamma A abundance.

In the low- and the high-NPP groups, we identified 39 and 20 data points of Gamma A abundance that were sampled in
cyclonic eddies, respectively, while more (204 and 55, respectively) were sampled in anticyclonic eddies. This is consistent
with the fact that eddies are more likely anticyclonic in the Northern Hemisphere, where our most (74%) sampling points were
located (Chelton et al., 2011).

The results showed that in the high-NPP group, the average Gamma A abundance within cyclonic eddies was one order of
magnitude higher than that in anticyclonic eddies or outside eddies (Fig. 7b), while the difference in the low-NPP group was
much smaller and statistically insignificant (t-test, $p > 0.05$) (Fig. 7a). The systematically higher Gamma A abundance is
unlikely to be caused by the locations of cyclonic eddies because most of the climatological factors were not significantly
different across types of eddies, except for a slightly lower dissolved Fe and DOC in cyclonic eddies in the high-NPP group
(t-test, $p < 0.05$) (Fig. S4a and S4c). We then further checked the residuals of the predicted Gamma A abundance using
climatological factors (i.e., Fig. 5c), still finding that the Gamma A abundance in cyclonic eddies in the high-NPP group was
significantly higher (one-tail t-test, $p < 0.05$) than the climatology-based predictions by on average a half order of magnitude,
while this was not the case for samples in anticyclonic eddies or outside eddies (Fig. 7d). Note that the residuals of predicted
Gamma A abundance in anticyclonic eddies in the low-NPP group were also significantly but only slightly higher than 0 (one-
tail t-test, $p < 0.05$) (Fig. 7c).


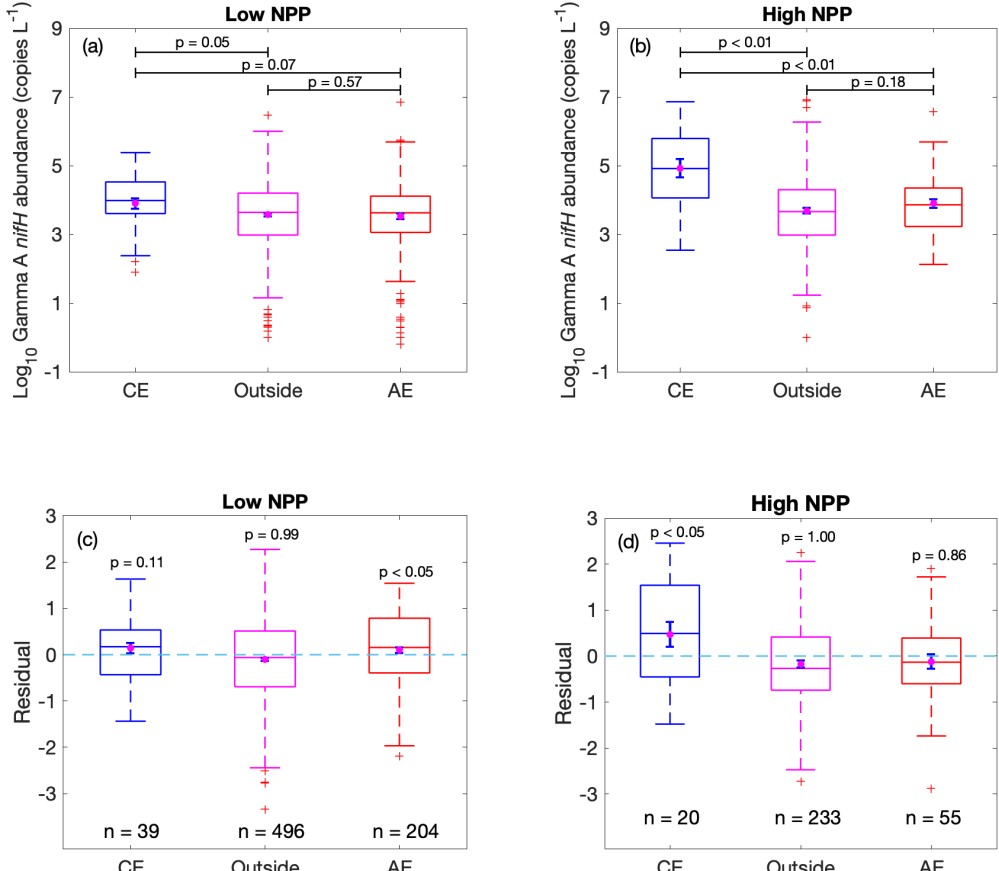

**Figure 7. Influence of mesoscale eddies on observed Gamma A abundance.** (a) Gamma A abundance and (b) residuals of predicted Gamma A abundance using climatological NPP and environmental factors in Fig. 5c grouped according to the data were sampled in cyclonic eddies (CE), anticyclonic eddies (AE) or outside eddies. The box plots show the median (central line), $25^{th}$ and $75^{th}$ percentile of data (upper and lower edges of box), $5^{th}$ and $95^{th}$ percentile (error lines) and outliers (red crosses). The error bars within boxes show the mean value (purple dots) and its standard error. Values above brackets are p-values of two tailed t-test whether the means of observed Gamma A abundance are equal (a–b) or one-tailed t-test whether the residuals are greater than zero (c–d).

These results indicated that cyclonic eddies could stimulate Gamma A abundance, but only in the high productivity oceans (> 400 mg C m$^{-2}$ d$^{-1}$ in this study). This finding is opposite to a previous hypothesis on autotrophic diazotrophs that anticyclonic eddies form a nitrate-depleted and well-lit environment favorable to N$_2$ fixation (Davis and Mcgillicuddy, 2006; Fong et al., 2008; Church et al., 2009; Liu et al., 2020). However, a sufficient supply of organic matter can play a prominent role in



heterotrophic $N_2$ fixation when the vertical pumping of nutrition-rich water driven by cyclonic eddies (Mcgillicuddy et al., 1998) can stimulate primary production (Falkowski et al., 1991). Nevertheless, the biogeochemical consequences of mesoscale eddies can be complex (Gaube et al., 2014; Mcgillicuddy Jr, 2016). For example, in addition to vertical pumping, the eddy stirring and trapping generated by mesoscale eddies can also have spatial effects on phytoplankton (Abraham, 1998; Wiebe
and Joyce, 1992). Further sampling and studies are still needed to improve our mechanistic understanding of the effects of mesoscale eddies on both autotrophic and heterotrophic $N_2$ fixation.

### 3.6 Reliability of Gamma A *nifH* data

It is questionable whether the *nifH* copies measured using qPCR and collected in this study can reliably represent the abundance of Gamma A or even NCDs in general. Previous studies found a large discrepancy between the PCR amplicon library and
qPCR copy number, and suggested that there existed preferential PCR amplification of Gamma A *nifH* genes (Shiozaki et al., 2017; Turk et al., 2011). As we already pointed out, the zero values of Gamma A *nifH* copies obtained from qPCR were very likely untrustworthy because they occurred more frequently than expected (Fig. S1). Although high Gamma A abundance over $10^6$ *nifH* copies $L^{-1}$ was observed in surface North Pacific Ocean, zero-value data were also massive (215 in total 608 data points) and even located close to those high-abundance data (Cheung et al., 2020) (Fig. S5a). It has also been found that
Gamma 4, whose primer was first designed in the Eastern South Pacific (Halm et al., 2012), might be a more versatile NCD phylotype in north Pacific Ocean (Cheung et al., 2021), although it is much less studied in other regions. Thus, whether preferential PCR amplification or qPCR detectability issue causes sampling bias in Gamma A abundance remains unknown and needs further study.

### 4. Summary and outlook

With more measurements becoming available, we explored in this study what factors controlled the distribution of a representative phylotype of heterotrophic diazotrophs, Gamma A, in the global ocean. Most of our findings imply that the supply of organic matter is the major determinant of Gamma A's abundance, confirming its heterotrophy. These findings include (1) the carrying capacity of Gamma A abundance increases with primary production and saturates at high primary production; (2) Gamma A benefits from high temperature probably because of the accelerated degradation rate of organic
matter; and (3) cyclonic eddies may stimulate the growth of Gamma A by introducing nutrients and elevating primary production. In addition, our analyses also revealed that Gamma A was more abundant in Fe-depleted areas, possibly to avoid competition with autotrophic diazotrophs in high-Fe environments. Overall, our study suggests that productivity and Fe can be factors differentiating niches between heterotrophic and autotrophic diazotrophs in the ocean, with the former favoring a high productivity and low-Fe niche, while the latter occupying the opposite.
However, the moderate explanatory power of our prediction model indicates that there must be other unknown factors and mechanisms also impacting heterotrophic diazotrophs. For instance, heterotrophic diazotrophs found in the guts of copepods (Scavotto et al., 2015) imply that they are subject to top-down controls, which was also suggested for marine autotrophic diazotrophs (Landolfi et al., 2021; Wang et al., 2019; Wang and Luo, in press). The uneven spatial samplings of Gamma A may also introduce biases into our analyses. Lastly, a universal primer is still lacking for detecting Gamma A and other NCDs

such as Alphaproteobacteria and Cluster III phylotype, which can also be important diazotrophs particularly in previously unrecognized regions for marine $N_2$ fixation (Wu et al., 2019; Langlois et al., 2008; Martínez-Pérez et al., 2018; Chen et al., 2019b). The combination of PCR amplification and metagenomic data can identify a broader NCD community (Delmont et al., 2018) and may help us design a better universal primer targeting major NCDs. More studies are needed in the future to improve our understandings of controlling factors, niches and distributions for heterotrophic diazotrophs, so that their

contribution to global marine $N_2$ fixation can be better evaluated.

**Data availability**

All the data used in this study are available in a data repository (https://doi.org/10.6084/m9.figshare.17284517) (Shao and Luo, 2021).

**Author contributions**

Y.-W.L. conceived and supervised the study. Y.-W.L. and Z.S. designed the study. Z.S. collected and analyzed the data and drafted the first version of the manuscript. Y.-W.L. and Z.S. contributed to the discussion of the results and revised the manuscript.

**Competing interests**

The authors declare that they have no conflict of interest.

**Acknowledgements**

The authors would like to thank the scientists and crew to sample the data that used in this study. The authors also thank Hua Wang, Yuhong Huang and Xiaoli Lu from XMU for their efforts in data collection and computation support. This study was funded by the National Natural Science Foundation of China (42076153 and 41890802).



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
