# Peer review of "Controlling factors on the global distribution of a representative marine non-cyanobacterial diazotroph phylotype (Gamma A)"

_Biogeosciences, 2021_

## Author Comment (AC1)

**Response to the comments from Anonymous Referee #2**

We thank Referee #2 for his/her efforts and provide very constructive comments that greatly helped us correct errors and improve the quality of our manuscript. We have responded (in blue fonts) to the comments point by point and revised the manuscript accordingly.

Shao and Luo attempt to better constrain the environmental drivers behind the observed biogeography of gamma A, a cosmopolitan marine non-cyanobacterial diazotroph group, using a metadata analysis of previously published gamma A abundances (estimated using qPCR targeting the gamma A nifH gene along with a suite of environmental parameters derived from the world ocean atlas, MODIS and several model outputs. On this whole, this represents a valid and interesting approach to gain insight into gamma A, but there are many items I feel need to be addressed prior to being considered for publication.

In general, I am concerned with ignoring all the apparent 0s in the compiled qPCR dataset. Better justification is needed for doing this – including why we should assume that abundance data would be normally distributed (in my experience with this type of data, it certainly isn't always), and better justification for the stated assumption that these undetects are not true 0s due to primer specificity. Is there any precedent for ignoring 0s in other published work that uses GAMs or other similar analyses?

Response: We thank the referee to bring up this very import issue. After carefully considering the referee's comments, we now agree that a large fraction of the zero-value data of Gamma A nifH copies were true zeros: The non-zero Gamma A abundance data were approximately log-normally distributed as shown in Fig. S1. Because the detection limits for nifH abundance usually ranges from $10^1$ to $10^2$ copies L$^{-1}$, the number of data that were not true zero but were below detection was very likely no more than 72, assuming the detection limit was $10^2$ copies L$^{-1}$ (Fig. S1). Therefore, the fact that there were far more zero data points (682) in our dataset indicated a large fraction of zero data could represent true absence of

Gamma A.

Additionally, based on the above analyses that many zeros represent true absence, we now also agree with the referee that the Gamma A is patch in space and time. The patchiness of diazotrophs, as suggested by the study recommended by the referee (Robidart et al., 2014), can be a consequence of lateral transport and mixing of water masses. The patchiness of Gamma A was also supported by the facts that many non-zero and zero Gamma A data were spatially close to each other (Fig. 1) and by our new analyses in the revised manuscript (new Fig. S2), showing that the environmental conditions of the non-zero and the zero Gamma A data largely overlapped.

In the revised manuscript, we still decided not to include zero-value data in the statistical analyses. The first reason was the patchiness of Gamma A distribution, which implicated that Gamma A can be either present or absent even when the environmental conditions we analyzed in this study were suitable for

Gamma A. That is, the presence of Gamma A needs a suitable environment, but a suitable environment does not necessarily guarantee the presence of Gamma A. If the zero-value data were included otherwise, similar environmental conditions could associate with both substantial and zero abundance, which would bias the response function of our statistical analyses, particularly as the fraction of zero-value data was large (~1/3) in our Gamma A dataset. Another reason was that we cannot identify true or false zeros of the Gamma A data, particularly considering the accuracy of qPCR that was highly sensitive to sample preservation, extraction protocol and the reliance of the standard curve (Smith and Osborn, 2009).

We found several marine ecological data analyses also removed zero-value abundance data and only used presence data (Irwin et al., 2012; Xiao et al., 2019). Their main reasons are similar: (1) reliability of zero data highly depends on the difficulty in species detection, and (2) large fraction of zeros would bias the response function of commonly used statistical analysis.

Therefore, we have revised the reasoning why the zero-value Gamma A abundance data were not included in the GAM (Method Section 2.1), added the description of the zero values and compared the environmental conditions associated to zero and non-zero data (Results Section 3.1), and revised discussion on the reliability of Gamma A *nifH* data (Section 3.6).

Method (Section 2.1):
"The non-zero abundance data was approximately log-normal distribution (Fig. S1). There were 682 data points reporting zero *nifH* copies which theoretically could indicate that Gamma A was either true absent or its abundance was below the detection limit in the samples. As the reported detection limit of qPCR usually ranges from $10^1$ to $10^2$ copies $L^{-1}$, the number of data that were below detection, according to log-normal distribution of observed non-zero data, was very likely no more than 72 even assuming a large detection limit of $10^2$ copies $L^{-1}$ (Fig. S1). The fact that there were far more zero data (682) in our dataset indicated a large fraction of zero data could represent true absence of Gamma A. Therefore, the distribution of Gamma A could be patchy, which was also confirmed by the mixed spatial distribution of the zero and non-zero data (see Results). The patchiness of diazotrophs in a small temporal and spatial scale has been widely found as a consequence of lateral transport and mixing of water masses (Robidart et al., 2014).
The patchiness of Gamma A implicated that it could be either present or absent even when the environmental conditions were suitable. That is, the presence of Gamma A needs a suitable environment, but a suitable environment does not necessarily guarantee the presence of Gamma A. Partly for this reason, the zero-value abundance data of Gamma A were not included in our further analyses. If the zero-value data were included otherwise, similar environmental conditions could associate with both substantial and zero abundance (Fig. S2), which would bias the response function of our statistical analyses, particularly as the fraction of the zero-abundance data was large (~1/3) in all the Gamma A data. Another reason why the zero-abundance data were not included was that, considering the accuracy of qPCR was highly sensitive to sample preservation, extraction protocol and the reliance of the standard curve (Smith and Osborn, 2009), it was difficult to identify whether the zero values represented true absence or below-detection abundance of Gamma A."

Results (Section 3.1):
"Although high Gamma A abundance over $10^6$ *nifH* copies $L^{-1}$ was observed in surface North Pacific Ocean, zero-value data were also massive (215 in total 608 data points) and even located close to those high-abundance data (Cheung et al., 2020) (Fig.1), indicating the patchy distribution of Gamma A. As discussed already, zero-abundance data were not included in the further analyses due to the patchiness of

Gamma A and the limitations of qPCR method in detecting true absence of Gamma A."

Results (Section 3.4.7):

"It was interesting that although Gamma A was undetected in the all samples in the South Pacific Gyre (Fig. 1) and all these zero-value data were not included in our GAM analyses, the prediction still showed the lowest Gamma A in this region (Fig. 6a), partly supporting the robustness of our prediction on Gamma A. However, another study suggested that NCDs were major players of $N_2$ fixation in SPG (Halm et al., 2012), which could reflect a possibility that Gamma A may not always be the dominant

NCD phylotype in the ocean. For example, Gamma 4 was suggested as a more versatile NCD phylotype in north Pacific Ocean (Cheung et al., 2021)."

Results (Section 3.6):

It is questionable whether the *nifH* copies measured using qPCR and collected in this study can reliably represent the abundance of Gamma A or even NCDs in general. When metadata are used, the reliability of comparison among absolute quantification can be affected by methodological factors. For example, even highly reproducible standard curves may result in significant variations in quantities of the same template in separated qPCR assays (Smith et al., 2006) due to the log nature of the curve. Extraction method of nucleic acid, sample preparation, variations in the efficiencies of the qPCR, differences in the qPCR platform can also impact the quantitative results (Smith et al., 2009). In addition, the copy numbers of *nifH* gene in Gamma A's genome remains unknown. There existed a large uncertainty that to what extend *nifH* gene copies can represent Gamma A abundance, especially in contrast to its autotrophic counterparts. All these problems will need better technology to resolve in the future.

Reference
Cheung, S. Y., Nitanai, R., Tsurumoto, C., Endo, H., Nakaoka, S., Cheah, W., Lorda, J. F., Xia, X. M., Liu, H. B., and Suzuki, K.: Physical forcing controls the basin-scale occurrence of nitrogen-fixing organisms in the North Pacific Ocean, Global Biogeochem Cy, 34, 9, https://doi.org/10.1029/2019GB006452, 2020.
Cheung, S., Zehr, J. P., Xia, X., Tsurumoto, C., Endo, H., Nakaoka, S.-i., Mak, W., Suzuki, K., and Liu, H.: Gamma4: a genetically versatile Gammaproteobacterial nifH phylotype that is widely distributed in the North Pacific Ocean, Environ. Microbiol., 23, 4246-4259, https://doi.org/10.1111/1462-2920.15604, 2021.

Halm, H., Lam, P., Ferdelman, T. G., Lavik, G., Dittmar, T., LaRoche, J., D'Hondt, S., and Kuypers, M. M. M.: Heterotrophic organisms dominate nitrogen fixation in the South Pacific Gyre, ISME J, 6, 1238-1249, https://doi.org/10.1038/ismej.2011.182, 2012.

Irwin, A. J., Nelles, A. M., and Finkel, Z. V.: Phytoplankton niches estimated from field data, Limnol Oceanogr, 57, 787-797, https://doi.org/10.4319/lo.2012.57.3.0787, 2012.

Robidart, J. C., Church, M. J., Ryan, J. P., Ascani, F., Wilson, S. T., Bombar, D., Marin, R., Richards, K. J., Karl, D. M., Scholin, C. A., and Zehr, J. P.: Ecogenomic sensor reveals controls on N2-fixing microorganisms in the North Pacific Ocean, ISME J, 8, 1175-1185, 10.1038/ismej.2013.244, 2014.

Smith, C. J., Nedwell, D. B., Dong, L. F., and Osborn, A. M.: Evaluation of quantitative polymerase chain reaction-based approaches for determining gene copy and gene transcript numbers in environmental samples, Environ. Microbiol., 8, 804-815, 2006.

Smith, C. J. and Osborn, A. M.: Advantages and limitations of quantitative PCR (Q-PCR)-based approaches in microbial ecology, FEMS Microbiology Ecology, 67, 6-20, 10.1111/j.1574-6941.2008.00629.x, 2009.

Xiao, W. P., Wang, L., Laws, E., Xie, Y. Y., Chen, J. X., Liu, X., Chen, B. Z., and Huang, B. Q.: Realized niches explain spatial gradients in seasonal abundance of phytoplankton groups in the South China Sea, Prog. Oceanogr., 162, 223-239, https://doi.org/10.1016/j.pocean.2018.03.008, 2018.

[Figure]

**Figure S2. Environmental conditions of observed Gamma A absence and presence data. Absence consists of zero data, UD (under detection) and ND (no detected) in dataset.**

[Figure]

**Figure 1.** Gamma A abundance including zero-value points (*nifH* copies L-1). The panels show data in depth ranges of (a) 0-25 m, (b) 25-100 m, (c) 100-200 m and (d) below 200 m. For clear demonstration, data are binned to 2° × 2° and geometric means in each bin are shown. Zero data were denoted as black triangles.

I also find much of the discussion to be speculative, esp. when trying to relate these findings to the broader group of gamma proteobacterial diazotrophs, or NCDs in their entirety.

Response: Thanks for your comment. We revised our manuscript and tried to remove speculative discussions (see more details below).

I suggest sticking with non-cyanobacterial diazotrophs throughout in place of "heterotrophic diazotrophs".

Response: Replacement has been done.

Line 7 – First sentence is awkward. Perhaps "non-cyanobacterial diazotrophs (NCDs) may be contributors to global marine...."

Response: We have rephrased the sentence as "Non-cyanobacterial diazotrophs (NCDs) may be contributors to global marine $N_2$ fixation, "

Line 10 – This needs definition since this is not a commonly used term for this sort of data. Is this even the right term to be used here and throughout? Aren't you really talking simply about abundance?

Response: Thanks for your comments. We represented Gamma A abundance using its *nifH* copies in this study. We will define this term in the abstract as "First, we represented Gamma A abundance by its *nifH* qPCR copies reported in literature, and analyzed its relationship to climatological biological and environmental conditions."

Line 15 – because the GLMs only explain some of the variance in gamma A abundances, I suggest using less definite terms here and throughout, e.g. in line 18 "most likely determined by" to "influenced by", etc.

Response: Thanks for your suggestion. We have found some place with definite term and revised them accordingly.

Line 75 "Our analyses revealed that local primary productivity, temperature, dissolved Fe concentration and the occurrence of cyclonic eddies can be the main factors impacting the distribution of Gamma A in the global ocean." to "Our analyses suggested that local primary productivity, temperature, dissolved Fe concentration and the occurrence of cyclonic eddies can be the main factors impacting the distribution of Gamma A in the global ocean."

Line 158 "**Primary production determines the maximal Gamma A abundance" to "Primary production supports the maximal Gamma A abundance"**

Line 170 "These results indicated that local NPP could largely determine the carrying capacity of Gamma A abundance, …" to "These results indicated that local NPP could largely support the maximal observed Gamma A abundance,…"

Line 262 "Our GAM results also revealed a positive relationship between silicate and $\Delta_{\text{Gamma-A}}$ in both the low- and the high-NPP groups (Figs. 4f and 4n)" to "Our GAM results also suggested a positive relationship between silicate and $\Delta_{\text{Gamma-A}}$ in both the low- and the high-NPP groups (Figs. 4f and 4n)"

Line 371 "In addition, our analyses also revealed that Gamma A was more abundant in Fe-depleted areas, possibly to avoid competition with autotrophic diazotrophs in high-Fe environments" to "In addition, our analyses also suggested that Gamma A was more abundant in Fe-depleted areas, possibly to avoid competition with autotrophic diazotrophs in high-Fe environments"

Line 17 – "mesoscale" in place of "short-term"

Response: Corrected.

Line 18 – "matter" in place of "matters

Response: Corrected.

– "provide insight into" in place of "insight a"

Response: Corrected.

Line 25 – remove heterotrophic here

Response: Corrected.

Line 26 – these aren't the best papers to cite here

Response: we have updated the citations here.

"non-cyanobacterial diazotrophs (NCDs) have been widely detected (e.g., Moisander et al., 2008; Langlois et al., 2008; Halm et al., 2012; Moisander et al., 2014; Shiozaki et al., 2014)"

Reference:

Moisander, P. H., Beinart, R. A., Voss, M., and Zehr, J. P.: Diversity and abundance of diazotrophic microorganisms in the South China Sea during intermonsoon, ISME J, 2, 954-967, https://doi.org/10.1038/ismej.2008.51, 2008.

Langlois, R. J., Hummer, D., and LaRoche, J.: Abundances and distributions of the dominant nifH
phylotypes in the Northern Atlantic Ocean, Appl Environ Microbiol, 74, 1922-1931, https://doi.org/10.1128/AEM.01720-07, 2008.

Halm, H., Lam, P., Ferdelman, T. G., Lavik, G., Dittmar, T., LaRoche, J., D'Hondt, S., and Kuypers, M. M. M.: Heterotrophic organisms dominate nitrogen fixation in the South Pacific Gyre, ISME J, 6, 1238-1249, https://doi.org/10.1038/ismej.2011.182, 2012.

Moisander, P. H., Serros, T., Paerl, R. W., Beinart, R. A., and Zehr, J. P.: Gammaproteobacterial diazotrophs and nifH gene expression in surface waters of the South Pacific Ocean, ISME J, 8, 1962-1973, https://doi.org/10.1038/ismej.2014.49, 2014.

Shiozaki, T., Ijichi, M., Kodama, T., Takeda, S., and Furuya, K.: Heterotrophic bacteria as major nitrogen fixers in the euphotic zone of the Indian Ocean, Global Biogeochem Cy, 28, 1096-1110,
https://doi.org/10.1002/2014gb004886, 2014.

Line 28 – "had higher relative abundances than" in place of "were far superior in number to"

Response: Corrected.

Line 30 – remove "dominant" or find a way to rephrase

Response: we have changed dominant to abundant. "Metagenomic studies also revealed the abundant
presence of diverse $N_2$-fixing proteobacteria in ocean genomic databases (Delmont et al., 2018; Delmont et al., 2021)."

Line 33 – rephrase "heavy"

Response: We have changed "heavy" to "significant"

Line 33-36 – Marine N2 fixation by NCDs is not quantified at all – please rephrase and make it
clear that there is only indirect evidence, including nifH transcription which does not "support" active N2 fixation by NCDs at all, it only provides another line of indirect evidence.

Response: Thanks for your comments. We have rephrased the sentence as "Although the $N_2$ fixed by non-cyanobacterial diazotrophs has not been quantified, substantial $N_2$ fixation found in aphotic zones (Rahav et al., 2013; Bonnet et al., 2013) and in experiments with photosynthetic inhibitors (Rahav et al., 2015;
Geisler et al., 2020), as well as recovered transcripts of the NCD *nifH* gene (Fernandez et al., 2011; Gradoville et al., 2017), provided a line of indirect evidence of non-cyanobacterial $N_2$ fixation in the ocean."

Line 36-37 – in addition to being uncultivated there are likely diverse niches and metabolic strategies used by this broad group. I might move the paragraph beginning at line 64 up, so that
you can make this point and introduce gamma A earlier.

Response: Thanks for your comments. We agree and have revised.

Line 38 – "Apparently" is awkward here – remove. Add "presumably" before "depending"

Response: Corrected.

Line 41 – this is misleading – there was no gamma A in Benavides et al. 2018, and they did not
assess other NCDs in this study

Response: Thank you for your comment. In Benavides et al. 2018b, abundant Gamma A *nifH* DNA copies were detected(Table 1of the paper). It was the expression of Gamma A *nifH* gene that was not detected in any DOM addition experiment (including controlled group). The authors also mentioned that the expression of Gamma A nifH was not detected "Despite being the most abundant ambient group as
determined by DNA qPCR counts" (page 6).
Therefore, we have rephrased this sentence as "However, DOM addition sometimes did not stimulate *nifH* expression of Gamma A even when its DNA copies was ambient (Benavides et al., 2018b), implying DOM may not always stimulate the activity of Gamma A.

Line 46 – Bonnet et al., citation makes no sense here.

Response: We meant that NCD may be like other cyanobacterial diazotrophs that they can provide N to diatoms. But we agree with the referee that this statement is too speculative. We then has revised the sentence as: "NCDs were also detected in diatom mats (Martínez et al., 1983), implying another novel habitat for NCDs".

Line 46 – "equip" is awkward

Response: Thanks for this comment. We have changed "equip with" to "contain".

Line 55 – is "supposably" needed here?

Response: Thanks for this comment. We have deleted "supposably" here.

Line 69 – as above, gamma A nifH transcription doesn't "reveal" it's important role in marine N2 fix

Response: Thanks for this comment. We have revised to "suggesting its role in marine $N_2$ fixation".

Line 71 – state here that this data is compiled from nifH-based qPCR studies.

Response: Thanks for this comment. In this study, we collected, to our best knowledge, all the reported in situ measurements of Gamma A nifH copies. We would restate this sentence as "With more data becoming available in the recent years, we collected, to our best knowledge, all the reported in situ
measurements of Gamma A *nifH* copies using qPCR assays, …"

Line 85 – zero nifH copies can also be true zeros

Response: Thanks for this comment. Please see our response to the general comments. We agree and have made necessary revisions.

Line 87 – all studies have different detection limits based on filter volumes, extraction volumes,
the amount of template used in the qPCR, etc. This is misleading.

Response: Thanks for this comment. Yes, We agree there is no common qPCR detection limit. Usually the detection limit ranges from $10^1$ to $10^2$ copies L$^{-1}$. We have revised to:

As the reported detection limit of qPCR usually ranges from $10^1$ to $10^2$ copies L$^{-1}$, the number of data that were below detection, according to log-normal distribution of observed non-zero data, was very likely no more than 72 even assuming a large detection limit of $10^2$ copies L$^{-1}$ (Fig. S1). The fact that there were far more zero data (682) in our dataset indicated a large fraction of zero data could represent true absence of Gamma A.

Please see our response to the general comments for more related details.

Line 95 – replace "were" with "have been" and you should note that these studies are specific to cyanobacterial diazos, and we do not know gene copy #s in gamma A.

Response: Thanks for this comment. We have revised the texts as follows:

"In the following analyses, we represented Gamma A abundance using its *nifH* copies, although we noted that variations in *nifH* copies in different cyanobacterial diazotroph cells have been reported (White et al., 2018; Sargent et al., 2016) and *nifH* copy numbers in Gamma A genome remain unknown.

Table 1 – There are additional studies represented in the Figure S5, it seems? These should be listed in Supp.

Response: Thanks for your comment. All the studies we used in our manuscripts have been listed in Table 1. Fig. S5 only included zero-value data comparing to Fig. 1. (In the revised manuscript, Fig.1 has also included zero-value data and Fig. S5 has been removed.)

Line 118 – More description needed about how cyclonic and anticyclonic eddies were called. What does a "clear shape" mean? Why is SLA missing from Table 2?

Response: Thanks for your comment. We define the core of mesoscale eddy as where the outermost closed contour line of the SLA field is. If a sampling point located in the eddy core, we recorded it as within anticyclonic eddy (positive SLA) or cyclonic eddy (negative SLA). We have revised this paragraph as:

"To identify if the Gamma A abundance was sampled in cyclonic or anticyclonic eddies, we extracted from AVISO program (www.aviso.altimetry.fr) the satellites-merged daily sea level anomaly (SLA) for the sampling days of the Gamma A data. The cores of mesoscale eddies were identified by the outermost closed contour lines of the SLA field. Only those sampling points located in cyclonic (negative SLA) and anticyclonic (positive SLA) eddies cores were recorded. Otherwise, data points were recorded as 'outside the eddy'."

We have also added data source of SLA in Table 2.

Line 148 – maybe specify it was undetected in this SPOT sample?

Response: Thanks for your comment. We have checked the reported data, Gamma A was detected but not quantified in this SPOT sample. The deepest datum was sampled at 1700m in South China Sea, but Gamma A nifH was undetected. Therefore, we have changed this sentence into:
"The deepest datum with detectable Gamma A nifH was sampled at 885 m in Southern California Bight (Hamersley et al., 2011)."

Line 150 – there are other studies that describe the depth distribution patterns of gamma A, eg. Chen et al., 2019, which seems to be missing from your list of studies???

Chen, Tien-Yi, et al. "Community and abundance of heterotrophic diazotrophs in the northern South China Sea: revealing the potential importance of a new alphaproteobacterium in N2 fixation." *Deep Sea Research Part I: Oceanographic Research Papers* 143 (2019): 104-114.

Response: Thanks for your comment. We have added the data from Chen et al. (2019) in our dataset and in fig. S2 (becoming Fig. S3 in revised manuscript). Also, we update this figure by including the zero-abundance data, which made us to revise the sentence here:

Available data showed that nifH abundance decreased with depth in the Southwestern Pacific Ocean, the Indian Ocean and the South China Sea, but did not have an apparent trend from the surface down to 200 m in the tropical Atlantic Ocean (Fig. S3).

Also revised accordingly in Section 3.4.6: "The decrease in Gamma A abundance with depth (Fig. S3 a, c and d; Moisander et al., 2008; Langlois et al., 2015; Chen et al., 2019b; Shiozakiet al., 2014; Wu et al., 2019) may therefore be attributed to" … "The nearly constant Gamma A abundance with depth in the Tropical Atlantic Ocean (Figs. S2b) can be the results of"…

[Figure]

**Figure S3**. Vertical profiles of Gamma A abundance in (a) Southwest Pacific Ocean, (b) tropical Atlantic Ocean and (c) India Ocean, (d) South China Sea. Dashed lines show all the sampled profiles, and solid lines and error bars are the mean and standard error in depth ranges of 0–10 m, 10–50 m, 50–100 m, 100–150 m and 150–250 m. Zero data was presented by 1 copy L$^{-1}$ in this figure.

Figure 2 caption – it's not clear why some data was singled out as "highest" and shown with red dots, while other high datapoints were left out – much better description needed.

Response: The caption was revised:

"**Figure 2. The relationship between Gamma A abundance and net primary production.** Both Gamma A abundance and net primary production (NPP) are log10-transformed. The data with NPP of $10^{2.0}$–$10^{2.6}$ mg C m$^{-2}$ d$^{-1}$ (the "low" NPP range) are divided into 6 groups with equal log-NPP intervals (i.e., divided at NPP of $10^{2.1}$, $10^{2.2}$, $10^{2.3}$, $10^{2.4}$ and $10^{2.5}$ mg C m$^{-2}$ d$^{-1}$), and the highest Gamma A abundance is identified in each group (red dots). The NPP-supported maximal Gamma A abundance (red line) is estimated by linearly fitting the red dots in the low NPP range, and saturates at $10^{7.0}$ *nifH* copies L$^{-1}$ for NPP > $10^{2.6}$ mg C m$^{-2}$ d$^{-1}$ (the "high" NPP range).

Section 3.2 and elsewhere – as above, I wonder whether carrying capacity is a needed term – at minimum it needs to be better defined, especially since the term has ecological ramifications that I am not sure are relevant here.

Response: The term "carrying capacity" was used in the previous manuscript to represent the maximal observed Gamma A abundance at given level of local net primary production. However, (as another referee also commented), the term "carrying capacity" has a strict ecological meaning. We decided to replace this term with ""NPP-supported maximal Gamma A abundance" in the revised manuscript.

Line 171 – "gamma A is expected to require a sufficient..."

Response: Corrected.

Line 208 – not clear why linear correlations are needed if the GAM is more reliable.

Response: The term "carrying capacity" was used in the previous manuscript to represent the maximal observed Gamma A abundance at given level of local net primary production. However, (as another 370 referee also commented), the term "carrying capacity" has a strict ecological meaning. We decided to replace this term with ""NPP-supported maximal Gamma A abundance" in the revised manuscript.

Line 219 – "is presumed to be" in place of "was supposably"

Response: Corrected.

Line 235 – too speculative

Response: Thanks for your comment. We accepted that this conclusion is too speculative. We have deleted this sentence in our manuscript.

Section 3.4.5 – although this relationship is interesting, this discussion is speculative, thus needs to be better phrased – e.g. interpreting this as "indirect" evidence supporting the hypothesis that 380 some NCDs are motile is misleading.

Response: Thanks for your comment. Our main hypothesis is Gamma A may benefit from the association with diatom. Swimming motility gene was suggested as a potential mechanism to find favorable niche and probably an indication of particle-attached lifestyle in Delmont et al. (2018). We have deleted this misleading message and revised this paragraph as:

"Our GAM results also suggested a positive relationship between silicate and $\Delta_{\text{Gamma-A}}$ in both the low- and the high-NPP groups (Figs. 4f and 4n), indicating a possible association between Gamma A and diatoms. NCDs have been found on the surface of diatoms  or on the diatom mats (Martínez et al.,

1983) as discussed above. Diatom-dominant ecosystems tend to produce abundant large particles either from dead diatoms and their aggregates or the fecal pellets generated by zooplankton (Tréguer et al., 2018). The large particles can be a good habitat for NCDs as already discussed. Our results then provide indirect evidence for the association between Gamma A and diatom."

Line 275 – Abundance does not equal active N2 fixation. No evidence that gamma A fixes anywhere, including the mesopelagic. Needs rewording.

Response: Thanks for your comment. Line 275 did not have the relevant message, and we guessed you were talking about the last sentence of this paragraph (line 278): …"can be the results of active transport of organic matter from the surface that fuels heterotrophic $N_2$ fixation in the dark deeper ocean."

We then changed "fuels heterotrophic $N_2$ fixation" to "supports the growth of Gamma A".

Line 292 – was this described in the methods? More details would be helpful.

Response: Thanks for your comment. This was not described in method section because we used this relationship based on the observations that the maximal observed Gamma A abundance increased with NPP (i.e. red line in Fig. 2). Considering the coherence and clarity of the paper, we think it will be better to describe this here rather than in the methods part.

We defined $\Delta_{Gamma-A}$ as the observed Gamma A abundance minus corresponding NPP-supported maximal observed Gamma A in logarithmic space, which practically removed the impact of NPP in $\Delta_{Gamma-A}$. Then, we analyzed other controlling factors on $\Delta_{Gamma-A}$ (mentioned in section 3.3) by using GAM. Therefore, predicted Gamma A abundance can be received by predicted $\Delta_{Gamma-A}$ plus the modeled NPP-supported maximal Gamma A abundance.

We have revised this sentence as:

"As described above, $\Delta_{Gamma-A}$ was defined as the Gamma A abundance minus corresponding NPP-supported maximal Gamma A abundance. After $\Delta_{Gamma-A}$ was predicted by controlling factors other than NPP using GAM (Figs. 5a-b), it was added back to the NPP-supported maximal Gamma A abundance (i.e., the red line in Fig. 2) to form a prediction model for the Gamma A abundance (Fig. 5c).

Line 300 – I would begin this discussion with an emphasis that the model predicts high abundances where gamma A is not observed, like the Southern Ocean and coastal areas.

Response: Thanks for your comment. We would rephrase this sentence as: "The results suggested that the Gamma A was most abundant in the Southern Ocean and the upwelling region in the Eastern Tropical South Pacific (Fig. 6A) where, however, Gamma A was not sampled (Fig. 1)." (Note that by reassessing we have decided to remove "coastal areas" from the sentence.)

Line 305 – remove "where"

Response: Corrected.

Line 354 - I'm confused why the PCR bias is mentioned here - there is no end-point PCR data included in this study. I think a more relevant discussion could include unknown copy #s in gamma A's genome, or even accuracy of qPCR in general, due to the reliance on standard curves.

Response: Thanks for your comment. We have deleted PCR bias here, and revised this paragraph substantially as listed above (in our response to general comments).

Line 359 - N2 fixers have been shown to be very patchy in space and time, see Robidart et al., 2014.

Response: Thanks for your comment. We agree that $N_2$ fixers can be very patchy (and therefore many
zeros are true zeros). We have added this argument in our manuscript (please see our response to general comments).

Line 367 – remove "confirming its heterotrophy" – over interpretation.

Response: Corrected.

Line 368 – replace "include" with "suggest" or the like

Response: Corrected.

Line 379 – there are many "universal" nifH primers with varying performance – do you mean a universal qPCR assay (which is unrealistic and would be difficult to interpret data from)?

Responses: Thanks for your comment. We agree that the statement is incorrect. What we wanted to express was that more NCD phylotypes were needed to be quantified, as Gamma A can only represent
part of gammaproteobacterial diazotrophs. Therefore, we have revised this sentence as:
"Lastly, future study should also consider qPCR primer and probe sets targeting other NCDs such as Alphaproteobacteria and Cluster III phylotype, which can also be important diazotrophs particularly in previously unrecognized regions for marine $N_2$ fixation (Wu et al., 2019; Langlois et al., 2008; Martínez-Pérez et al., 2018; Chen et al., 2019b)."

---

## Author Comment (AC2)

**Response to the comments from Anonymous Referee #1**

We thank Referee #1 for his/her very thorough and constructive comments that helped us greatly improve our manuscript. We have responded (in blue fonts) to the comments point by point and revised the manuscript accordingly.

Shao and Luo compile published abundance data of Gamma A (qPCR nifH gene counts), a putatively heterotrophic diazotroph widespread in the oceans. Using ancillary data, atlas, satellite products and models, they perform a thorough statistical analyses to infer relationships between Gamma A and environmental variables. Their results suggest that Gamma A benefits from primary production by-products and is mostly dominant in warm and iron-poor waters of the ocean. The data analyses are extensive and the results worth publishing.

However, the authors should improve the comparison between their study and Langlois et al. 2015, who also compiled Gamma A data and performed statistical analyses to define their niche. How does the present study build up on previous ones?

Response: Thanks for your comments. Although Langlois et al. (2015) has been referred in multiply places in the previous version of the manuscript, we agree with the reviewer that we should revise the manuscript so that our results can be better compared to Langlois et al. (2015).

In Langlois et al. (2015),  the authors statistically analyzed the relationship between the Gamma A nifH abundance and a suit of environmental parameters including nutrients, salinity, temperature and oxygen. They found Gamma A was mostly distributed in the warm (tropical) and oligotrophic surface. With more data becoming available in the recent years, we used 80% more (1795 vs 992 data points) Gamma A *nifH* abundance data in our study than those in Langlois et al. (2015). We also used five additional variables including primary production, iron and DOC concentrations, solar radiation and mixed layer depth, as well as submesoscale eddies, to more thoroughly analyze potential controlling factors on Gamma A. Part of our conclusions is consistent to Langlois et al. (2015) that Gamma A prefers warm environment. But our study has revealed that Gamma A also prefer high primary production and cyclonic eddies, suggesting that sufficient supply of organic matter can be the more important determinant of Gamma A distribution.

We have revised the manuscript as follows:

(1) The last paragraph of 1. Introduction:
"Langlois et al. (2015) analyzed the distribution of Gamma A phylotype in the Pacific and Atlantic Oceans and suggested that Gamma A preferred warm and oligotrophic surface oceans. With more data becoming available in the recent years, we collected, to our best knowledge, all the reported in situ measurements of Gamma A *nifH* copies using qPCR assays, compiling a dataset with 1795 data points, 80% more than those used in Langlois et al. (2015). We then analyzed the relationship between this *nifH*-based Gamma A abundance and the long-term background of ecological and environmental factors by using their climatological monthly averages. To more thoroughly analyze potential controlling factors on Gamma A, we included 5 variables, including primary production, iron and DOC concentrations, solar radiation and mixed layer depth, in addition to temperature and concentrations of nitrate, phosphate and silicate that used in Langlois et al. (2015). We further explored the influence of mesoscale eddies on Gamma A abundance."

(2) First paragraph of 4. Summary and outlook:
… in the global ocean. "The results of our study did not agree with the conclusion of a previous study that Gamma A preferred oligotrophic oceans (Langlois et al. 2015). Instead, most of our findings" …

I also found several mis-citations, where the wrong citations are given to justify a statement or where the message of a given paper was not well understood.
In all, the exercise seems statistically correct but of relatively poor ecological interpretation significance unless several points are improved. Below I provide a list of comments.

Response: Thank you very much for your comments. We have responded to all specific comments point by point below.

Specific comments

L7: Delete "the" in "to the global marine".

Response: Corrected.

L10: What is the carrying capacity? This term is used throughout the manuscript whereas it is never really
explained.

Response: The term "carrying capacity" was used in the previous manuscript to represent the maximal observed Gamma A abundance at given level of local net primary production. However, (as another referee also commented), the term "carrying capacity" has a strict ecological meaning. We decided to
replace this term with "'NPP-supported maximal Gamma A abundance" in the revised manuscript.

L15: "in addition" not "in additional".

Response: Corrected.

L17: Eddies are not short-term features, they may last several months. Please choose another term.

Response: We have changed "short-term features" to "mesoscale features".
L18: "organic matter" not "organic matters"

Response: Corrected.

L19: Weird wording, please rephrase.

Response: We have rephrased that "and therefore provide an insight into niche differentiation between the heterotrophic and autotrophic $N_2$ fixation."

L20: "sampling" not "samplings", and delete "better" from the end of the sentence.

Response: Corrected.

L32: "oxygen deficient zones"

Response: We have corrected "oxygen deplete zones" to "oxygen deficient zones".

L33: "heavy" sounds weird, please rephrase.

Response: We have switched "heavy" to "significant".

L33: I would not say heterotrophic $N_2$ fixation is "not well quantified", it's just not quantified at all. There is -currently- no assay able to isolate heterotrophic $N_2$ fixation from autotrophic $N_2$ fixation.

Response: Thanks for your comment. We have revised the sentence as "Although the $N_2$ fixed by NCDs has not been quantified, …".

L41: This is not what these papers really say. In Benavides 2018b Gamma A was not detected, so it does not necessarily mean it was not stimulated by DOM, it just was not present in the samples
at all. In Benavides 2015 $N_2$ fixation in dark waters was stimulated by amino acids.

Response: Thank you for your comment. In Benavides et al. 2018b, abundant Gamma A *nifH* DNA copies were detected (Table 1of the paper). It was the expression of Gamma A *nifH* gene that was not detected in any DOM addition experiment (including controlled group). The authors also mentioned that the
expression of Gamma A nifH was not detected "Despite being the most abundant ambient group as determined by DNA qPCR counts" (page 6).
The referee was correct that amino acids did stimulate $N_2$ fixation in dark waters in Benavides 2015, while the effect of sugar was not obvious.
Therefore, we have rephrased this sentence as "However, DOM addition sometimes did not stimulate
*nifH* expression of Gamma A even when its DNA copies was ambient (Benavides et al., 2018b), implying DOM may not always stimulate the activity of Gamma A. In addition, the response of aphotic $N_2$ fixation to different DOM composition could also vary (Benavides et al., 2015)."

L42-43: NCDs are thought to be attached to particles, but they haven't been found to be attached to particles.

Response: Thanks for this comment. We have changed "attach to particles" to "associate with particles".

L45: Bonnet 2016 find that N released by Trichodesmium is taken up by diatoms. Not N released by NCDs.

Response: We meant that NCD may be like other cyanobacterial diazotrophs that they can provide N to diatoms. But we agree with the referee that this statement is too speculative. We then has revised the sentence as: "NCDs were also detected in diatom mats (Martínez et al., 1983), implying another novel
habitat for NCDs".

 L45: "to equip" sounds weird.

Response: We have changed "to equip with" to "contain".

L52-53: How did those studies look at DIN inhibition of individual NCDs strains? It seems this is
not what these studies really did.

Response: Shown in FIG 6 of Bentzon-Tilia et al. (2015), $N_2$ fixation in NCD strains *P. stutzeri* BAL361 and *R. ornithinolytica* BAL286 decreased significantly when $NH_4^+$ was added. However, in another NCD strain R. palustris BAL398, $N_2$ fixation increased dramatically upon the addition of $NH_4^+$, which they
speculated was because the nitrogenase complex had a function in addition to N acquisition, such as using $N_2$ fixation as an electron sink for $NH_4^+$ consumption.
The inhibition of $NH_4^+$ to NCD strain *S. castanea* was also observed in Martínez-Pérez et al. (2018).

L58: This is quite unfair to say, please cite:

Bombar, Deniz, Ryan W. Paerl, and Lasse Riemann. 2016. "Marine Non-Cyanobacterial Diazotrophs: Moving beyond Molecular Detection." Trends in Microbiology 24 (11): 916–27.

Cornejo-Castillo, Francisco M., and Jonathan P. Zehr. 2020. "Intriguing Size Distribution of the Uncultured and Globally Widespread Marine Non-Cyanobacterial Diazotroph Gamma-A." The ISME
Journal. https://doi.org/10.1038/s41396-020-00765-1.

Langlois, Rebecca, Tobias Großkopf, Matthew Mills, Shigenobu Takeda, and Julie LaRoche. 2015. "Widespread Distribution and Expression of Gamma A (UMB), an Uncultured, Diazotrophic, γ-Proteobacterial NifH Phylotype." PloS One 10 (6): 1–17.

Moisander, Pia H., Mar Benavides, Sophie Bonnet, Ilana Berman-Frank, Angelicque E. White, and
Lasse Riemann. 2017. "Chasing after Non-Cyanobacterial Nitrogen Fixation in Marine Pelagic
Environments." Frontiers in Microbiology. https://doi.org/10.3389/fmicb.2017.01736.

Riemann, Lasse, Hanna Farnelid, and Grieg F. Steward. 2010. "Nitrogenase Genes in Non-
Cyanobacterial Plankton: Prevalence, Diversity and Regulation in Marine Waters." Aquatic Microbial
Ecology: International Journal 61 (3): 235–47.

Response: Thanks for your suggestion. In Bombar et al. (2016), the discussed controlling factors were
presence of oxygen, presence of reactive inorganic nitrogen and the availability of energy.. Iron was
mentioned as an important component of nitrogenase but not its effect on NCDs. Langlois et al. (2015)
mentioned Gamma A may rely on DOC accumulated in the upper water column due to vertical
stratification. In Cornejo-Castillo et al. (2020), the controlling factor discussed is the abundance of
Gamma A in different size fractions. Moisander et al. (2017) reviewed studies related to NCDs and the
major controlling factor discussed is DIN and DOM. Riemann et al., 2010 also discussed the impact of
carbon, nitrogen and oxygen on NCDs. These studies did not directly analyzed the relationship between
NCDs and iron/stratification.

Therefore, we have revised the texts as:

"Regarding other important factors that control autotrophic diazotrophs, iron (Fe) may potentially impact
NCDs if they also depend on the high Fe-containing nitrogenases to fix $N_2$ (Bombar et al. 2016), although,
as discussed above, the $N_2$ fixation by NCDs is still not quantified. Strong stratification may also benefit
NCDs by accumulating organic matter in the upper water column (Langlois et al. 2015). However, there
have been, to our knowledge, no studies analyzing the effects of Fe or stratification on NCDs.

L59: change "can" to "may".

Response: Corrected.

L63: You may cite Benavides, M., and J. Robidart. 2020. "Bridging the Spatiotemporal Gap in
Diazotroph Activity and Diversity With High-Resolution Measurements." Frontiers in Marine
Science 7. https://doi.org/10.3389/fmars.2020.568876.

Response: We have added this citation.

L69: "suggesting" would be fairer than "revealing" here. Note that nif genes can be used for other
purposes.

Response: Thanks for your suggestion. We have made this correction.

L84: "the upper 100 m of the water column".

Response: Corrected.

L87: there is no common qPCR detection limit, it depends on the essay, the machine, the lab, the volume of water filtered.

Response:

The referee was correct that there was no common qPCR detection limit. Usually, the detection limit ranges from $10^1$ to $10^2$ copies L$^{-1}$. The number of presence data points (72) under detection can be considered overestimated when the larger detection limit ($10^2$ copies L$^{-1}$) was taken into calculation if the data can be assumed normally distributed and all the zero-value data present low abundance below the detection limit.

However, we have removed the sentence in the revised manuscript upon the comments from Referee #2. We now agree with Referee #2 that some reported zero-value data were true zeros and the distribution of Gamma A can be patchy. We have added a new subsection in Results to discuss the zero-value data.

Table 1: I am a bit puzzled at 0 m depths, this is unlikely. Please check.

Response: Thanks for your reminding. We rechecked the original data reported in published papers, depths of 0 m were included. We supposed this may represent that the data were sampled at sea surface.

L103-104: I wonder why an artificial neural network was considered for DOC concentrations when there is a now a global database available https://odv.awi.de/data/ocean/dom-compilation-hansell-et-al-2021/ Please reconsider using it instead.

Response: Thanks for your suggestion. Most our Gamma A samplings do not have DOC data available
in the Hansell's global database sampled in the same spatial and month grids. Indeed, the DOC data we used is produced from an artificial neural network model based on the same DOC observation database mentioned by the referee.

L116: It is unclear how SLA data was used, data was extracted from the same days as Gamma A samples were taken? Please explain.

Response: Thanks for your comment. Yes, SLA data was extracted from the same days as Gamma A samples were taken. We define the core of mesoscale eddy as where the outermost closed contour line of the SLA field is. If a sampling point located in the eddy core, we recorded it as within anticyclonic eddy (positive SLA) or cyclonic eddy (negative SLA). We have revised this paragraph as:

"To identify if the Gamma A abundance was sampled in cyclonic or anticyclonic eddies, we extracted from AVISO program (www.aviso.altimetry.fr) the satellites-merged daily sea level anomaly (SLA) for the sampling days of the Gamma A data. The cores of mesoscale eddies were identified by the outermost closed contour lines of the SLA field. Only those sampling points located in cyclonic (negative SLA) and anticyclonic (positive SLA) eddies cores were recorded. Otherwise, data points were recorded as 'outside 240 the eddy'."

L149: nifH abundance also decreases with depth in the North Pacific (see work from Church at station ALOHA).

Response: Thanks for your comment. Yes, Church et al's work found nifH abundance decreases with depth in the North Pacific. However, their study mainly focused on cyanobacterial diazotrophs and did not report *nifH* qPCR copies of Gamma A.

L170: please explain what the carrying capacity is.

Response: As mentioned above, we have changed "carrying capactiy" to 'NPP-supported maximal Gamma A abundance' in our manuscript.

L172: Please cite Bombar 2016.

Response: We have added this citation.

L175: How are biogeographic patterns biased by the sparse and uneven sampling in different 260 ocean regions? Can this be assessed statistically?

Response: Thanks for your comments. Yes, spatial biases in samples existed in our data set. To partly eliminate this bias caused by concentrated samplings in specific regions, in the previous version we have binned our data points into 2° × 2° grids. In addition, the standard errors estimated by GAM can also help 265 to assess this kind of bias. Regions with undersampled biogeographic features would contain large uncertainties shown in Figure 6b. From our result, the largest uncertainties for the predictions exist in the Southern Ocean (Fig. 6b) because there were no Gamma A samples in this high-nitrate area. Other than this region, the uncertainty in the predicted Gamma A nifH abundance was at similar level in the global ocean (Fig. 6b), partly indicating that the spatial biases in samples may not impact our analyses greatly. 270 We have added a discussion of this issue at the end of the revised manuscript: "The samples of Gamma A nifH copies are still limited particularly in the Southern Hemisphere, possibly causing spatial biases in our analyses. More sampling and studies are needed in the future to improve our understandings" …

L190-191: This negative correlation is because low temperature anticorrelates with NPP, right?

Response: The response variable $\Delta_{\text{Gamma-A}}$ used in the analyses here was the "residual" of observed Gamma A abundance to the NPP-supported maximal Gamma A abundance , which therefore practically removed the effect of NPP.

Nevertheless, we agree with both referees' comments on the necessity of the univariate linear correlation analysis, and have decided to delete this section in the revised manuscript.

Section 3.4. The first sentence belongs in the methods. Why even show linear regressions at all if
the model is deemed better? I would suggest just mentioning the correlations, maybe move them to the supplementary, and dive into the GAM directly in the main text. Why are, in any case, the effects found using GAM so different to the ones obtained with linear correlations? (e.g. L219).

Response: Thanks for your comments. Univariate analysis was used in linear correlation while
multivariate analysis was used in GAM. Therefore, effect of every controlling factor in GAM is partial effect when other controlling factors were controlled. That is why effects found using GAM were different to those in linear correlation. Again, upon both referee's comments, we have decided to delete the linear correlations section (and associated discussions in other places) in our manuscript.

L223: fuel with what?

Response: Particulate organic matter (POM) can fuel Gamma A with organics. We have revised the sentence that "Lastly, particulate organic matter (POM) can also supply necessary organic carbon and nutrients to Gamma A …"
L224: I suggest replacing Farnelid 2019 for Riemann 2010.

Response: Thanks for your suggestion, we have made this replacement.

L235: This seems quite a speculative conclusion to make. DOC concentrations alone do not inform about lability, and, to date, we don't know anything about the metabolism of Gamma A or which kind of DOM molecules they may use.

Response: Thanks for your comment. We accepted that this conclusion is too speculative. We have
deleted this sentence in our manuscript.

L243-244: note that NCDs also need P.

Response: Thanks for your comment. P* in our study represents excess inorganic phosphate in seawaters. NCD's source of P remains unknown. We have revised our manuscript to: "while our results tentatively indicate that the competition may not occur strongly between NCDs and phytoplankton, although it is still unclear whether NCDs use inorganic or organic P sources."

    L301: this seems quite different from observations.

Response: Thanks for your comment. First, we agree that the statement of high Gamma A abundance in coastal regions can be misleading and should be removed. Second, Gamma A nifH copies were under sampled in the Southern Ocean and the upwelling region in the Eastern Tropical South Pacific (fig. 1), therefore we actually do not have direct supporting measurements. In the rest of the paragraph, we then discussed the largest uncertainties associated the high predicted Gamma A abundance in the Southern

Ocean. The high Gamma A abundance predicted in the upwelling region in the Eastern Tropical South Pacific was mostly generated by its high NPP and temperature (Fig. S3 b and f).

    We then have revised the manuscript as follows:

    "The results suggested that the Gamma A was most abundant in the the Southern Ocean and the upwelling region in the Eastern Tropical South Pacific (ETSP) (Fig. 6A) where, however, Gamma A was not sampled (Fig. 1). The predicted high abundance in the Southern Ocean was mostly caused by its high nitrate concentration (Figs. S3g–h). However, the largest uncertainties for the predictions also exist in the Southern Ocean (Fig. 6b) as there were no Gamma A samples in this high-nitrate area (Fig. 1). The predicted high abundance in ETSP resulted from its simultaneously high NPP and temperature (Fig S3 a, b, e and f). Although the direct measurements of Gamma A nifH copies also lacked in ETSP, the sufficient samples in other regions with high NPP and temperature lowers the uncertainties of the predicted high abundance of Gamma A in ETSP (Fig. 6b). Nevertheless, future sampling in these two regions can then test our predictions and reduce the uncertainties."

Figure 6: why is the abundance "annual"? it's not a rate.

    Response: Thanks for your comment. Annual mean abundance represents the mean value of Gamma A abundance from January to December. This term has been used often in other studies related to global distribution of species (e.g., Flombaum et al., 2013; Li, 1998). We then decided to keep this term

    Reference:
    Flombaum, P., Gallegos, J. L., Gordillo, R. A., Rincón, J., Zabala, L. L., Jiao, N., ... & Martiny, A. C. (2013). Present and future global distributions of the marine Cyanobacteria Prochlorococcus and Synechococcus. Proceedings of the National Academy of Sciences, 110(24), 9824-9829. (Fig. 2)

    Li, W. K. (1998). Annual average abundance of heterotrophic bacteria and Synechococcus in surface ocean waters. Limnology and oceanography, 43(7), 1746-1753.

    Section 3.5. the connection or justification of why the effect of SLA is tested here is hard to follow.

Response: Thanks for your comment. For the environmental factors analyzed above, we used their climatological monthly values, which certainly may depart from in situ conditions (note that most of our Gamma A samples did not have sufficient in situ environmental parameters reported). Mesoscale eddies are one kind of phenomena that causes the in situ conditions different systematically from the climatological conditions. Also, as we mentioned in Introduction, mesoscale eddies can influence nitrogen fixation in the ocean. Therefore, we wanted to discover whether the Gamma A abundance in these eddies were systematically different from those predicted by our GAM model using climatological environmental conditions.

We have revised the first paragraph of Section 3.5 as

"The root-mean-square error (RMSE) of 0.86 and an $R^2$ of 41% in the prediction model (Fig. 5c) indicated that there was still substantial unexplained variance in Gamma A abundance. One possible reason was that we used the climatological monthly means in the environmental factors, while the in situ conditions can differ greatly from the climatological values. For example, oceanic mesoscale eddies can influence biogeochemical process not only by advective transport but also by variations in the biological and chemical environments (McGillicuddy, 2016). Particularly, as discussed above, some regional studies have suggested that mesoscale eddies may influence the distribution of autotrophic diazotrophs and/or NCDs. We then explored whether the occurrence of mesoscale eddies can impact Gamma A abundance."

McGillicuddy Jr, D. J.: Mechanisms of physical-biological-biogeochemical interaction at the oceanic mesoscale, Ann Rev Mar Sci, 8, 125-159, https://doi.org/10.1146/annurev-marine-010814-015606, 2016.

L316: eddies are not short term phenomena.

Response: Thanks for your comment. We have changed short term phenomenon to mesoscale phenomenon.

320: but also the number of data points in the NH is much higher than in the SH, potential bias, how can it be assessed?

Response: Thanks for your comment. Yes, the biases existed. However, this kind of biases is hard to be assessed. Similar to the biases caused by uneven sampling discussed above, we believed more samplings in the South Hemisphere are needed to reduce the bias (which has been emphasized in the last paragraph of the revised manuscript).

L367: "confirming heterotrophy" seems quite risky. We need genomic data and tracer experiments to confirm that.

Responses: Thanks for your comment. We have deleted "confirming its heterotrophy".

L379: Unclear here, nifH primers are universal. There is no primer for cyanobacterial diazotrophs only. These primers target all diazotrophs with Mo nitrogenases.

Responses: Thanks for your comment. We agree that the statement is incorrect. What we wanted to express was that more NCD phylotypes were needed to be quantified, as Gamma A can only represent part of gammaproteobacterial diazotrophs. Therefore, we have revised this sentence as:

"Lastly, future study should also consider qPCR primer and probe sets targeting other NCDs such as Alphaproteobacteria and Cluster III phylotype, which can also be important diazotrophs particularly in previously unrecognized regions for marine $N_2$ fixation (Wu et al., 2019; Langlois et al., 2008; Martínez-Pérez et al., 2018; Chen et al., 2019b)."

---

## Author Response (AR2)

**Response to the comments from Anonymous Referee #2**

Thanks to Shao and Luo for thoughtful responses to both reviewers' comments. Two of my original comments were not addressed fully, so I raise them here again.

Response: Thank you again for your helpful comment. Please kindly see our responses to these two comments below.

First and most importantly, in the Figure 2 caption (with implications elsewhere). Although more description is supplied, I am still unclear why the high data points at 10^2.0 were left out of the linear regression. Please provide justification.

Response: We apologize for this confusion. We now have included the data point at NPP of $10^2$ in the linear regression. In addition, partly suggested by another reviewer, we have shortened the intervals of $Log_{10}$NPP to 0.05 when identifying the maximal observed Gamma A abundance, thus having more data points for a more reliable linear regression between the highest observed Gamma A abundance and NPP (see Fig. 2).

Secondly, it still seems that there is data in the global plots (now only fig. 2) that is not cited in Table 1 from the Eastern South Pacific (at a minimum). Please triple check that all data is properly cited.

Response: Although all the Gamma A abundance data shown in Fig. 1 in the previous version had been cited in Table 1 (the data in the the Eastern South Pacific were from Shiozaki et al. (2018a)), as suggested by the reviewer, we rechecked all of our data sources and found two additional papers that also reported undetected (i.e. zero-value) Gamma A abundance in the Eastern South Pacific (Halm et al., 2012; Turk-Kubo et al., 2014). (It was missed in the previous version probably because Gamma A were reported using an uncommon acronym "AO") We have added these new zero-value data in Fig. 1 and cited the papers in Table 1.

Halm, H., Lam, P., Ferdelman, T. G., Lavik, G., Dittmar, T., LaRoche, J., . . . Kuypers, M. M. M. (2012). Heterotrophic organisms dominate nitrogen fixation in the South Pacific Gyre. The ISME Journal, 6(6), 1238-1249. doi:https://doi.org/10.1038/ismej.2011.182

Shiozaki, T., Bombar, D., Riemann, L., Sato, M., Hashihama, F., Kodama, T., . . . Furuya, K. (2018a). Linkage between dinitrogen fixation and primary production in the oligotrophic South Pacific Ocean. Global Biogeochemical Cycles, 32(7), 1028-1044. doi:https://doi.org/10.1029/2017gb005869

Turk-Kubo, K. A., Karamchandani, M., Capone, D. G., & Zehr, J. P. (2014). The paradox of marine heterotrophic nitrogen fixation: abundances of heterotrophic diazotrophs do not account for nitrogen fixation rates in the Eastern Tropical South Pacific. Environ Microbiol, 16(10), 3095-3114. doi:https://doi.org/10.1111/1462-2920.12346

**Response to the comments from Anonymous Referee #3**

This is an interesting paper displaying the putative controlling factors of Gamma A, the most sampled non-cyanobacterial diazotroph (NCD) in the ocean. However, I noticed outstanding over-
speculations throughout the manuscript, in which some of the discussions were even baseless. Although the previous reviewers had also pointed out the same problem, it seems that the authors insisted on their speculations.

Response: We thank the reviewer for the very constructive comments that have greatly improved our manuscript. After carefully reading the reviewer's general and specific comments, we now have modified our analysis and revised the manuscript to fully avoid those speculations (in particular, the speculation that Gamma A is heterotrophy and uses organic matter from primary producers). In the previous version, we assumed that Gamma A was supported by NPP, and therefore the dependent variable of the GAM
analysis was the reduction (residual) of observed Gamma A abundance from the "NPP-supported maximal abundance". In the present version, we have discarded this method. In the GAM analysis, the variable to be predicted is now the Gamma A abundance itself, and NPP is added to the predictors with other environmental variables. The GAM analysis now applies to the entire dataset, not to two separated groups according to NPP as was done in the previous version. This new GAM analysis revealed some
different features from the previous analysis. We believe that the new GAM analysis is more objective and that the results are more robust. For example, after directly adding NPP into GAM as one of the predictors, the GAM reached a higher explanatory power while not identifying a substantial relationship between Gamma A abundance and temperature or DOC. We would thank the reviewer again for her/his very useful comments that make the analysis more solid.

Also, I have some concerns regarding their approaches. I doubt if the authors should use monthly climatological factors as the predictors of Gamma A, given that the ocean is highly dynamic and the diazotrophs are usually patchily distributed. More details please see the specific comment.

Response: We completely agree with the reviewer that the ocean is highly dynamic and that the real environment can be different from climatological conditions. That was also one of the reasons why we also analyzed the relationship between mesoscale eddies and Gamma A abundance as an example to show that the dynamic ocean can also influence Gamma A. This is discussed in the first paragraph of 3.4: "The root-mean-square error (RMSE) of 0.84 and an $R^2$ of 43% in the prediction model (Fig. 4c) indicated that there was still substantial unexplained variance in Gamma A abundance. One possible reason was that we used the climatological monthly means for the environmental factors, while the in situ conditions can differ greatly from the climatological values. For example, oceanic mesoscale eddies can influence biogeochemical processes not only by ..."

Many climatological data of biogeochemical properties are available only in monthly intervals, particularly the nutrients. Therefore, we cannot conduct our analysis using data with shorter temporal resolutions. Nevertheless, the monthly climatological factors can also provide a large-scale background for understanding the general habitats of organisms. The practice of using monthly climatological data in meta-analysis appears common in marine ecology, including those for diazotrophs (e.g., Tang and Cassar, 2019). In a previous analysis of ocean $N_2$ fixation rates, our group even used yearly average climatology data as preditors (but with less data points) (Luo et al. 2014).

Tang, W., and N. Cassar (2019), Data-driven modeling of the distribution of diazotrophs in the global ocean, *Geophys. Res. Lett.*, 46, 12258-12269, doi:10.1029/2019gl084376.

Luo, Y.-W., I. D. Lima, D. M. Karl, C. A. Deutsch, and S. C. Doney (2014), Data-based assessment of environmental controls on global marine nitrogen fixation, *Biogeosciences,* 11(3), 691-708, doi:10.5194/bg-11-691-2014.

Regarding the patchness of diazotrophs, it was also raised by a reviewer in the first round and we have fully addressed this issue (fourth paragraph in Section 2.1), largely by only analyzing non-zero Gamma A abundance data. The reviewer who asked this question appeared to agree with our modification.

L8: Why did the authors presume that the NCDs are heterotrophs? In particular, the physiology of Gamma A is basically unknown. I suggest deleting "are presumably heterotrophic bacteria".

Response: Thank the reviewer for this comment. We stated that NCDs "are presumably heterotrophic bacteria" from the suggestions of other reviewers and also based on statements from other papers (Bombar et al., 2016; Zehr and Capone, 2020). However, we now agree with the reviewer that NCD should not be assumed to be heterotrophy (see response to general comments) and have deleted "are presumably heterotrophic bacteria" from the abstract.

Bombar, D., Paerl, R. W., and Riemann, L.: Marine non-cyanobacterial diazotrophs: moving beyond molecular detection, Trends Microbiol., 24, 916-927, https://doi.org/10.1016/j.tim.2016.07.002, 2016.

Zehr, J. P. and Capone, D. G.: Changing perspectives in marine nitrogen fixation, Science, 368, 729-+, https://doi.org/10.1126/science.aay9514, 2020.

L13-15: I am skeptical about the relationship between Gamma A abundances and the NPP estimated by remote sensing… Would it be caused by the relationships between GammaA and other factors like temperature or chlorophyll? Also, "NPP-supported maximal abundance" is highly speculative, which sounds like that you have already proven the direct relationship between Gamma A and NPP. It should be noted that correlation may not imply causation. Also, let's say the positive relationship between Gamma A and NPP is true, you cannot tell if Gamma A contributed to NPP or were supported by NPP…

Response: We understand that NPP from satellite remote sensing is derived from temperature and chlorophyll using certain algorithms, which however is beyond the scope of this paper. As a reference, in our modified multivariate GAM analysis, NPP and temperature, together with other environmental parameters, are added into potential predictors and the results suggest that Gamma A abundance has a generally positive relationship with NPP, but no clear pattern between Gamma A and temperature is generated particularly for temperature > 15 degreeC in which most Gamma A data were reported (Fig. 4g). In other words, Gamma A may be more related to NPP than temperature.

Regarding the "NPP-supported maximal abundance", we completely agree with the reviewer (see our more response of this issue to the general comments) and the term "NPP-supported maximal abundance" has been removed from the abstract and the entire paper.

L16-17: These interpretations are highly speculative, which should be avoid in the abstract.

Response: Thank you for this comment. We have deleted these interpretations.

L19: I think your result only tells that GammaA abundances were higher in the waters with higher
SLA, while it is uncertain whether these higher SLA values really mean eddies…

Response: We identified the cores of mesoscale eddies by the outermost closed contour lines of the SLA field which has been mentioned in the Methods section (below Eq. 2). We also slightly revised the text to make the statement clearer: "The cores of mesoscale eddies were identified by the outermost closed
contour lines of the SLA field. Only those sampling points located in the cores of cyclonic (negative SLA) or anticyclonic (positive SLA) eddies were recorded.".

L20: How were Gamma A affected by the organic matters? It needs to be more specific.

Response: As we no longer assume Gamma A is heterotrophy, this sentence has been removed.

L21: It is not true. Gamma A are positively correlated with temperature, just like other cyanobacterial diazotrophs.

Response: We tried to state that some (not all) predictors for Gamma A were different from those for autotrophic diazotrophs. We have revised the abstract with our new results and made the statement more precise: " Overall, our results suggest that Gamma A tends to inhabit ocean environments with high productivity and low iron concentrations, and therefore provide insight into the niche differentiation of Gamma A from cyanobacterial diazotrophs, which are generally most active in oligotrophic ocean regions and need a sufficient iron supply, although both groups prefer well-lit surface waters. "

L55: Gamma A were not detected in aphotic waters.

Response: Thank you for the comment. This sentence has been deleted in the revised manuscript.

L63: Yes, some NCDs could be autotrophic/mixotrophic, and that's why it is inappropriate to presume that NCDs are heterotrophs. Also, as Gamma A are generally more abundant in surface water, they could be photoheterotrophic or even phototrophic.

Response: We thank the reviewer for mentioning this. As stated in the response to general comments, we have avoided the speculation that Gamma A is heterotrophy. Additionally, our new results revealed that Gamma A tended to habitat well-lit waters, which was discussed in the revised manuscript.

L66: Please specific what "active" means here. Were the NCDs active in fixing nitrogen in these habitats?

Response: We changed to "substantial presence of NCDs are found in DIN-replete environment …"

L126-132: We don't know the real suitable conditions for Gamma A yet… This problem could be simply because the currently available/commonly used environmental data does not cover the real controlling factors of Gamma A.

Response: Thanks for this comment. We agreed and revised the text to incorporate this comment: "It can also indicate our limited understandings of environmental conditions: The currently available environmental data do not include all the controlling factors of Gamma A."

L154: The resolution of monthly averaged environmental data seems too low to predict the abundances of Gamma A.

Response: Please see our response to the general comments for the same question.

L230-L235: What will the correlation look like if you calculate the correlation coefficients using all the data points? Also, the amount of data points in "high NPP" and "low NPP regions" are largely different, which should be considered.

Response: This correlation stated here (0.21) was calculated using all the data points. The number of data points in "low NPP" is two times greater than that in "high NPP". As stated in our response to the general comments, the dataset was no longer separated into low-NPP and high-NPP groups in the GAM analysis in the revised manuscript.

L245: We don't know if Gamma A need organic matters from primary producers. It is simply your speculation from your observation, and you cannot use your speculation to explain your observation…

Response: We agree with the comment. We have revised this paragraph entirely to more objectively and logically evaluate the implication of the positive correlation between Gamma A and NPP:

" If the presumption that Gamma A is heterotrophic or photoheterotrophic bacteria (Bombar et al., 2016; Zehr and Capone, 2020) is true, a positive relationship between the Gamma A abundance and net primary production (NPP) can be expected because its energetically intensive $N_2$ fixation can benefit from a sufficient supply of organic matter from primary producers. The significant positive correlation between the logarithm of Gamma A nifH abundance and the logarithm of NPP in our data (correlation = 0.21, p < 0.01) (Fig. 2) was consistent with this presumption. However, this positive correlation could just reveal a fact that Gamma A and primary producers share certain common controlling factors. For example, even if Gamma A would be autotrophic or mixotrophic and can harvest energy from solar radiation, it could also positively correlate with NPP, as both of them would be supported by high light intensity. Although the capability of Gamma A to fix $N_2$ has not been quantified, it could also be possible that the fixed N by Gamma A, if it occurred, could in turn support NPP."

L255: The statement about the "NPP-supported maximal Gamma A abundance" is baseless. As we don't even know about the trophic status of Gamma A.

Response: It has been removed.

L286: GAM can delineate partial effects of different variables, including NPP. I am wondering why the authors artificially separate the dataset based on the NPP? Analysing the full dataset with GAM may result in more universal conclusions about the putative determinants of Gamma A. Also, I doubt that the linear regression based on 6 data points (in Fig. 2) means any valid relationship between Gamma A and NPP.

Response: Thank you for this comment. We have modified our analysis following this comment by including NPP in GAM and no longer separating the dataset based on NPP in GAM (see our response to the general comments).
In the revised manuscript, the linear regression line is an estimate just showing an approximate upper bound of the Gamma A abundance and is not used in future analysis (see our response to the general comments). Nevertheless, we shorten the intervals to pick more (12) data points for the linear regression.

L307: The discussion about DOC is high-speculative, which is fully depending on what the authors believe in… Based on the result of GAM, the correlations between DOC and Gamma A are contradictory with the authors' speculations about the NPP…

Response: DOC is no longer a significant predictor after we modified our analysis by including NPP as one of the GAM predictors and not separating high- and low-NPP groups (see our response to general comments). Therefore, the discussion for DOC has been removed entirely.